# LLM-Assisted Semantically Diverse Teammate Generation for Efficient Multi-agent Coordination

**Lihe Li** [1 2] **Lei Yuan** [1 2 3] **Pengsen Liu** [1 2] **Tao Jiang** [1 2 3] **Yang Yu** [1 2 3]

## Abstract

Training with diverse teammates is the key for learning generalizable agents. Typical approaches aim to generate diverse teammates by utilizing techniques like randomization, designing regularization terms, or reducing policy compatibility, etc. However, such teammates lack semantic information, resulting in inefficient teammate generation and poor adaptability of the agents. To tackle these challenges, we propose Semantically Diverse Teammate Generation (SEMDIV), a novel framework leveraging the capabilities of large language models (LLMs) to discover and learn diverse coordination behaviors at the semantic level. In each iteration, SEMDIV first generates a novel coordination behavior described in natural language, then translates it into a reward function to train a teammate policy. Once the policy is verified to be meaningful, novel, and aligned with the behavior, the agents train a policy for coordination. Through this iterative process, SEMDIV efficiently generates a diverse set of semantically grounded teammates, enabling agents to develop specialized policies, and select the most suitable ones through language-based reasoning to adapt to unseen teammates. Experiments show that SEMDIV generates teammates covering a wide range of coordination behaviors, including those unreachable by baseline methods. Evaluation across four MARL environments, each with five unseen representative teammates, demonstrates SEMDIV's superior coordination and adaptability. Our code is available at https://github.com/lilh76/SemDiv.

[1]National Key Laboratory for Novel Software Technology, Nanjing University [2]School of Artificial Intelligence, Nanjing University [3]Polixir Technologies. Correspondence to: Lei Yuan <yuanl@lamda.nju.edu.cn>, Yang Yu <yuy@nju.edu.cn>.

*Proceedings of the 42nd International Conference on Machine Learning*, Vancouver, Canada. PMLR 267, 2025. Copyright 2025 by the author(s).

## 1. Introduction

Recently, cooperative multi-agent reinforcement learning (MARL) has gained significant attention (Oroojlooy & Hajinezhad, 2023), demonstrating promising applications in various fields such as autonomous driving (Zhang et al., 2024c), domain calibration (Jiang et al., 2024), and financial trading (Huang et al., 2024). Classic MARL approaches (Lowe et al., 2017; Rashid et al., 2018; Wang et al., 2021; Yu et al., 2022) primarily focus on training a group of agents to cooperatively complete specific tasks and evaluate their performance in the same setting. However, in open multi-agent environments (Yuan et al., 2023b), agents are often required to team up with unseen teammates exhibiting diverse coordination behaviors. For instance, autonomous driving agents frequently encounter human drivers with a wide range of driving behaviors. In such scenarios, agents trained using conventional MARL techniques may struggle to coordinate effectively, as they tend to overfit to the behaviors of their training teammates.

Training with diverse teammates is the key for learning generalizable MARL agents. To generate diverse teammates, recent research in areas such as ad-hoc teamwork (Mirsky et al., 2022) and zero-shot coordination (Treutlein et al., 2021) has emerged. FCP (Strouse et al., 2021) trains teammates using different random seeds, while TrajeDi (Lupu et al., 2021) and MEP (Zhao et al., 2023) introduce diversity regularization terms for teammates. Other methods like LIPO (Charakorn et al., 2023), Macop (Yuan et al., 2023a), BRDiv (Rahman et al., 2023), and L-BRDiv (Rahman et al., 2024) induce diversity by reducing compatibility among teammates or between teammates and agents. While achieving some progress, these approaches primarily focus on policy-level diversity, generating teammates that lack semantic information and are not grounded into specific coordination behaviors. This limitation results in two significant challenges. First, the exploration of the teammate policy space is inefficient, as teammates are driven to optimize for differences at the policy-level rather than actively discovering novel coordination behaviors at the semantic-level. Second, agents are unable to utilize semantic information, and limited to trial-and-error interactions for teammate adaptation, hindering their deployment in costly tasks.

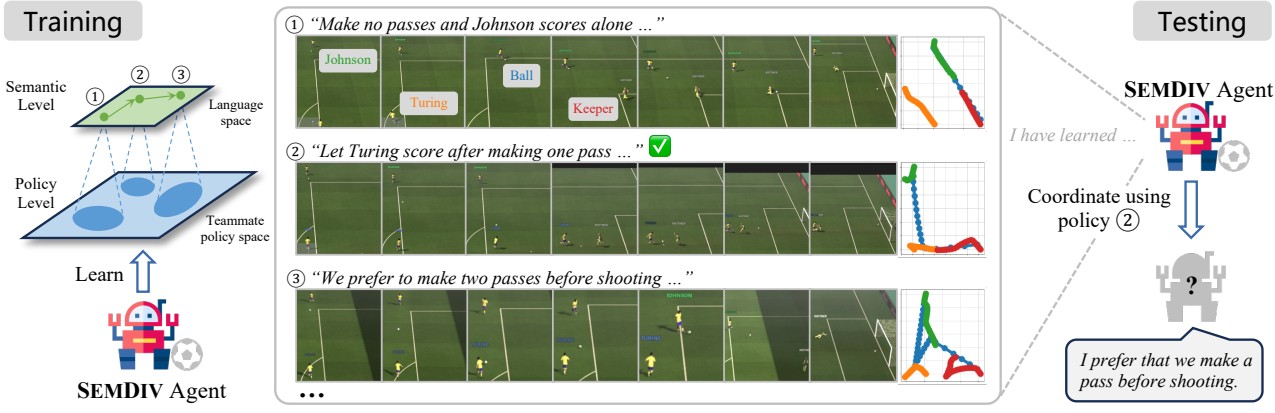

*Figure 1.* An overview of the training and testing process of SEMDIV. Left: During training, SEMDIV proposes novel coordination behaviors in natural language and transform them into teammate policies for agent learning. Right: During testing, SEMDIV takes as input the description of the unseen teammates and selects the optimal learned policy for coordination.

To tackle these challenges, we propose Semantically Diverse Teammate Generation (SEMDIV), a novel framework leveraging the capabilities of large language models (LLMs) to discover and learn diverse coordination behaviors at the semantic level, as illustrated in Figure 1. SEMDIV employs an iterative process: in each iteration, it first generates a novel coordination behavior described in natural language, then translates it into a reward function (Xie et al., 2024; Ma et al., 2024a) to train a teammate policy. Once the policy is verified to be capable of completing the task, distinct from previous teammates, and aligned with the behavior, the agents with multi-head architecture (Kessler et al., 2022; Yuan et al., 2024) train a new policy head for coordination. Through this process, SEMDIV efficiently generates a diverse set of semantically grounded teammates, enabling agents to develop specialized policies, and select the most suitable ones through language-based reasoning to adapt to unseen teammates with specific coordination behaviors.

We conduct experiments across four MARL environments, including Level-Based Foraging (LBF) (Papoudakis et al., 2021), Predator-Prey (PP) (Lowe et al., 2017), StarCraft Multi-Agent Challenge-v2 (SMACv2) (Ellis et al., 2023), and Google Research Football (GRF) (Kurach et al., 2020). SEMDIV successfully generates teammates with novel coordination behaviors unreachable by policy-level baselines, for example, multiple passes in GRF. Teaming up with five unseen teammates with distinct and representative coordination behaviors in each of the four environments, SEMDIV's agents outperform the best baseline by 19% for task success rate and 39% for the success rate of satisfying the teammates preferred coordination behaviors. These results highlight the capability of SEMDIV to train adaptive agents with strong coordination ability in open multi-agent environments.

## 2. Problem Formulation

In this work, we focus on cooperative MARL tasks where agents need to coordinate with unseen and uncontrollable teammates. This problem can be formulated as a tuple $\mathcal{M} = \langle \mathcal{N} = \mathcal{N}_{\mathrm{ag}} \cup \mathcal{N}_{\mathrm{tm}}, \mathcal{S}, \mathcal{A}, P, \Omega, O, R, \gamma \rangle$ by extending the Dec-POMDP framework (Oliehoek & Amato, 2016). Here, $\mathcal{N}$ is the set of all agents, divided into controllable agents $\mathcal{N}_{\mathrm{ag}} = \{1, \ldots, n_{\mathrm{ag}}\}$ and uncontrollable teammates $\mathcal{N}_{\mathrm{tm}} = \{n_{\mathrm{ag}} + 1, \ldots, n_{\mathrm{ag}} + n_{\mathrm{tm}}\}$. $\mathcal{S}$ is the set of global states, $\mathcal{A} = \mathcal{A}_{\mathrm{ag}} \times \mathcal{A}_{\mathrm{tm}} = \prod_{j \in \mathcal{N}_{\mathrm{ag}}} \mathcal{A}^j \times \prod_{k \in \mathcal{N}_{\mathrm{tm}}} \mathcal{A}^k$ is the joint action space. $P : \mathcal{S} \times \mathcal{A} \to \mathrm{Pr}(\mathcal{S})$ is the transition function, $\Omega$ is the set of observations, $O : \mathcal{S} \times \mathcal{N} \to \Omega$ is the observation function, $R : \mathcal{S} \times \mathcal{A} \times \mathcal{S} \to \mathbb{R}$ is the reward function, and $\gamma \in [0, 1)$ is the discount factor. At each time step $t$, agent $i \in \mathcal{N}$ receives an observation $o_t^i = O(s_t, i) \in \Omega$ and outputs an action $a_t^i \in \mathcal{A}^i$ with policy $\pi^i(\cdot|o^i)$. The joint action $\boldsymbol{a}_t = (a_t^1, \ldots, a_t^{n_{\mathrm{ag}} + n_{\mathrm{tm}}})$ leads to the next state $s_{t+1} \sim P(\cdot|s_t, \boldsymbol{a}_t)$ and a team reward $R(s_t, \boldsymbol{a}_t, s_{t+1})$. The objective of the controllable agents is to find a joint policy $\boldsymbol{\pi}^{\mathrm{ag}}(\cdot|\boldsymbol{o}^{\mathrm{ag}}) = \prod_{j \in \mathcal{N}_{\mathrm{ag}}} \pi^j(\cdot|o^j)$ that maximizes the expected return with unknown teammates $\boldsymbol{\pi}^{\mathrm{tm}} = \prod_{k \in \mathcal{N}_{\mathrm{tm}}} \pi^k$, i.e., $\mathbb{E}_{\boldsymbol{\pi}^{\mathrm{tm}}}[J(\boldsymbol{\pi}^{\mathrm{ag}}, \boldsymbol{\pi}^{\mathrm{tm}})] = \mathbb{E}_{\boldsymbol{\pi}^{\mathrm{tm}}}[\mathbb{E}_{s_t, \boldsymbol{a}_t}[\sum_t \gamma^t R(s_t, \boldsymbol{a}_t, s_{t+1})]]$.

As we aim to study teammate generation and agents coordination at the semantic-level, we consider scenarios in which the group of teammates[1] $\boldsymbol{\pi}^{\mathrm{tm}}$ provides a natural language description $b$ prior to the execution phase. This description outlines their preferred coordination behaviors, such as a specific plan to complete the task, or the occurrence of a particular coordination event, etc. The agents can leverage this natural language description $b$ to adapt their individual policies $\pi^{j \in \mathcal{N}_{\mathrm{ag}}}$, thereby aligning their actions with the co-

---

[1] For simplicity, we denote a group of teammates as "a teammate" hereafter when no ambiguity arises.

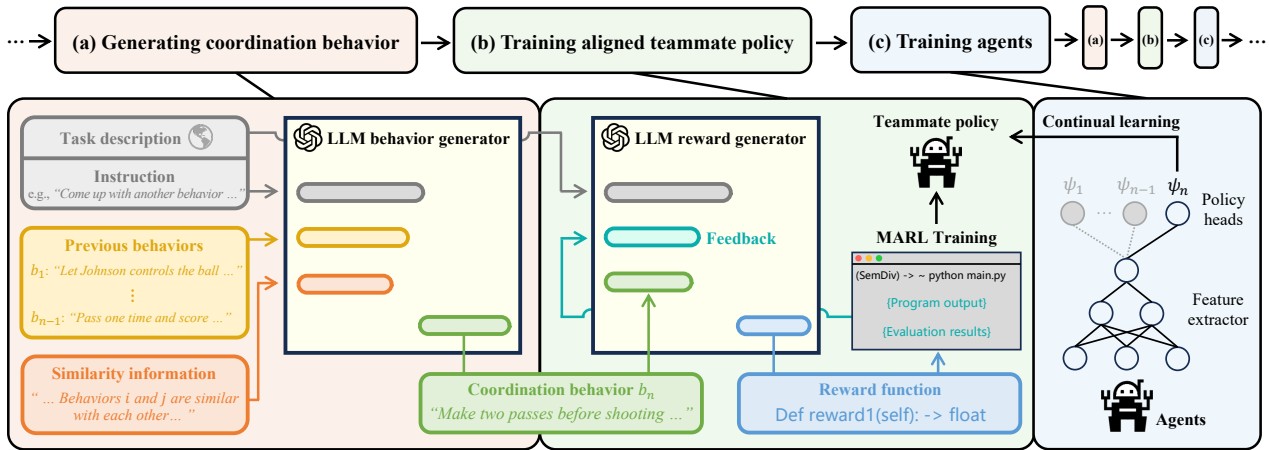

*Figure 2.* The overall workflow of SEMDIV. (a) Generating coordination behavior. SEMDIV iteratively generates of semantically diverse coordination behaviors, enabling efficient exploration of the teammate policy space. (b) Training aligned teammate policy. For each coordination behavior described in natural language, a teammate policy is trained to align with that behavior. (c) Training agents. Agents are continually trained with these teammates, developing strong coordination ability.

ordination preferences of $\pi^{\text{tm}}$, ultimately enhancing overall team coordination and task performance.

## 3. Method

This section introduces SEMDIV (Figure 2), a novel framework that leverages LLMs to efficiently generate semantically diverse teammates, and train agents with strong coordination ability. SEMDIV begins with the iterative generation of semantically diverse coordination behaviors, enabling efficient exploration of the teammate policy space (Section 3.1). For each coordination behavior described in natural language, a teammate policy is trained to align with that behavior (Section 3.2). Simultaneously, agents are continually trained with these teammates, enabling them to develop strong coordination ability and adapt efficiently to unseen teammates during execution (Section 3.3).

### 3.1. Iterative Generation of Semantically Diverse Coordination Behaviors

To derive semantically diverse teammates in a cooperative MARL task, SEMDIV first leverages an LLM to iteratively generate a diverse set of plausible coordination behaviors described in natural language.

Concretely, let $\mathcal{P}_{n-1} = \{(b_m, \pi_m^{\text{tm}}, I_m)\}_{m=1}^{n-1}$ denote the set of teammates generated in the previous $n-1$ iterations, where each tuple $(b_m, \pi_m^{\text{tm}}, I_m)$ consists of a behavior $b_m$, its corresponding policy $\pi_m^{\text{tm}}$, and a boolean value $I_m$ indicating whether the teammate is valid ($I_m = \texttt{True}$) or not ($I_m = \texttt{False}$). In the $n^{\text{th}}$ iteration, the LLM behavior generator takes a task description $\texttt{desc}$ and an instruction $\texttt{inst}$ as prompts. The description $\texttt{desc}$ includes the basic information about the environment, the agents, and the task

they need to complete. The instruction $\texttt{inst}$ is a simple sentence like "*come up with a possible and concrete coordination behavior*". When $n > 1$, to ensure novelty and diversity in each iteration, the prompt also includes previous behaviors $\mathcal{B} = \{b_m \in \{b_1, \ldots, b_{n-1}\} \mid I_m = \texttt{True}\}$, with explicit instructions in $\texttt{inst}$ for the LLM to avoid replicating these behaviors while proposing a new one. Furthermore, to ensure meaningful diversity in the generated teammates, SEMDIV incorporates a feedback mechanism to refine the behavior generation process. Specifically, when a pair of policies $\pi_m^{\text{tm}}, \pi_{m'\neq m}^{\text{tm}} \in \mathcal{P}_{n-1}$ are similar with each other, this information $\texttt{info\_sim}$ is fed back into the LLM prompt. For example, in a navigation task, different behaviors such as "move to point A" and "move to coordinate $(3, 4)$" might produce similar policies if point A is close to $(3, 4)$. By identifying such redundancies, a process elaborated later, the LLM gains a deeper understanding of the coordination task. This grounding feedback enables SEMDIV to iteratively generate coordination behavior-policy pairs that are diverse at both semantic and policy levels, enhancing exploration of the policy space. The full prompts for the LLM behavior generator are in Appendix F.2.

Next, the LLM behavior generator utilizes the prompt $p = [\texttt{desc}, \texttt{inst}, \mathcal{B}, \texttt{info\_sim}]$, along with its internal knowledge, to output a new concrete behavior $b_n$ in natural language. This behavior is then used to generate a corresponding policy $\pi_n^{\text{tm}}$. If $\pi_n^{\text{tm}}$ demonstrates the intended behavior $b_n$, is different from previous policies in $\mathcal{P}_{n-1}$, and completes the task, $I_n$ is set to $\texttt{True}$. Otherwise, $I_n$ is set to $\texttt{False}$. Then, $\mathcal{P}_n = \mathcal{P}_{n-1} \cup \{(b_n, \pi_n^{\text{tm}}, I_n)\}$. This iterative process continues until a sufficient number of valid teammates are generated, fostering the development of agents with strong coordination capabilities.

### 3.2. Grounded Generation of Each Single Teammate

This section describes how SEMDIV generates a teammate policy that aligns with a specified coordination behavior and completes the MARL task, while ensuring that the teammate policy is distinct from previously generated ones.

**Prompts to Reward Functions**  Within each iteration, given a coordination behavior $b_m$, SEMDIV uses an LLM to generate a corresponding reward function $\hat{R}_m : \mathcal{S} \times \mathcal{A} \times \mathcal{S} \to \mathbb{R}$ as an executable program. Similar to the behavior generator, the LLM reward generator takes the task description, an instruction, and feedback information as prompts. The task description must include basic callable attributes and APIs to ground the reward function in the task environment. For instance, in a 3D navigation task, attributes like `agent1_position: np.ndarray[(3,)]` and APIs like distance calculation functions should be provided. The instruction is a sentence like "write a reward function that formats as 'def reward(self) → float' and aligns with the coordination behavior $\{b_m\}$". However, with only the task description and instruction, the generated reward may not be able to train a valid teammate for several issues: (i) The reward function is not executable, e.g., it calls an undefined attribute. (ii) The teammate fails to complete the task after training with this reward function. (iii) The return of the reward function remains nearly constant during training, indicating that it's non-functional. (iv) The teammate does not demonstrate the intended coordination behavior $b_m$. (v) The teammate is similar to previously generated ones.

To address these issues, SEMDIV incorporates the above critical grounding feedback into subsequent prompts to iteratively refine the reward function. This iterative process continues until either a valid teammate policy $\boldsymbol{\pi}_m^{\text{tm}}$ is learned or the maximum number of attempts is reached. A valid policy is one that satisfies all verification criteria (described below), at which point the tuple $(b_m, \boldsymbol{\pi}_m^{\text{tm}}, \texttt{True})$ is added to $\mathcal{P}_{m-1}$. If the maximum number of attempts is reached, $(b_m, \boldsymbol{\pi}_m^{\text{tm}} = \texttt{null}, \texttt{False})$ is added instead. The prompts for this LLM reward generator are in Appendix F.3.

**Reward Functions to Policies**  Given an executable reward function $\hat{R}_m$, SEMDIV incorporates it into the environment code and leverages an off-the-shelf cooperative MARL algorithm to train the teammate policy $\boldsymbol{\pi}_m^{\text{tm}}$. The training objective is to maximize the self-play return defined as:

$$J(\tilde{\boldsymbol{\pi}}_m^{\text{tm}}, \boldsymbol{\pi}_m^{\text{tm}}) = \mathbb{E}_{s_t, \boldsymbol{a}_t} \left[ \sum_t \gamma^t \left( \lambda_1 r_t + \lambda_2 \hat{r}_t^m \right) \right], \quad (1)$$

where $\tilde{\boldsymbol{\pi}}_m^{\text{tm}}$ is the complementary policy of $\boldsymbol{\pi}_m^{\text{tm}}$, which controls agents $\mathcal{N}_{\text{ag}}$. It outputs actions $(a_t^1, ..., a_t^{n_{\text{ag}}})$, which are combined with the actions $(a_t^{n_{\text{ag}}+1}, \ldots, a_t^{n_{\text{ag}}+n_{\text{tm}}})$ output by $\boldsymbol{\pi}_m^{\text{tm}}$ to form the joint action $\boldsymbol{a}_t$. Rewards are com-

puted as the sum of two components: the task-specific reward $r_t = R(s_t, \boldsymbol{a}_t, s_{t+1})$ and the generated reward $\hat{r}_t^m = \hat{R}_m(s_t, \boldsymbol{a}_t, s_{t+1})$. For the weighting factors, $\lambda_1 = 1$, $\lambda_2$ decays from 1 to 0 over the course of training. This decay ensures that $\boldsymbol{\pi}_m^{\text{tm}}$ learns to complete the task.

**Policy Verification**  After training $\boldsymbol{\pi}_m^{\text{tm}}$, SEMDIV verifies its validity. First, it evaluates $(\tilde{\boldsymbol{\pi}}_m^{\text{tm}}, \boldsymbol{\pi}_m^{\text{tm}})$ for multiple episodes to compute returns for $r_t$ and $\hat{r}_t^m$, checking issues (ii) failure to complete the task, and (iii) non-functional rewards. For issue (iv), SEMDIV extracts the main information in these episodes, transforms it into natural language, and uses an LLM to confirm that $\boldsymbol{\pi}_m^{\text{tm}}$ demonstrates the intended coordination behavior $b_m$. For issue (v), we assume a joint agent policy $\boldsymbol{\pi}^{\text{ag}}$ that can effectively coordinate with all previous teammates $\Pi_{m-1} = \{\boldsymbol{\pi}_j^{\text{tm}} \in \mathcal{P}_{m-1} \mid I_j = \texttt{True}\}$, which will be elaborated in the next section. To confirm that $\boldsymbol{\pi}_m^{\text{tm}}$ is distinct from $\Pi_{m-1}$, we follow (Charakorn et al., 2023) and check whether the following condition holds:

$$\frac{J(\boldsymbol{\pi}^{\text{ag}}, \boldsymbol{\pi}_j^{\text{tm}}) - J(\boldsymbol{\pi}^{\text{ag}}, \boldsymbol{\pi}_m^{\text{tm}})}{|J(\boldsymbol{\pi}^{\text{ag}}, \boldsymbol{\pi}_j^{\text{tm}})|} > \epsilon, \quad (2)$$

for all $\boldsymbol{\pi}_j^{\text{tm}} \in \Pi_{m-1}$, under configurations $\lambda_1 = 1, \lambda_2 = 0$ and $\lambda_1 = 0, \lambda_2 = 1$, where $\epsilon > 0$ is a predefined threshold. If this condition is satisfied, $\boldsymbol{\pi}_m^{\text{tm}}$ is confirmed to be distinct, as $\boldsymbol{\pi}^{\text{ag}}$ cannot effectively coordinate with it. Otherwise, similarity information is recorded and provided as feedback to the LLM behavior generator, as described in Section 3.1. This verification process ensures the quality and diversity of each generated teammate. The prompts used for behavior-policy alignment verification are detailed in Appendix F.4.

### 3.3. Continual Learning and Execution of the Coordinating Agents

The goal of SEMDIV is to derive a joint agent policy $\boldsymbol{\pi}^{\text{ag}}$ that can effectively coordinate with both self-generated and unseen teammates based on natural language descriptions of their coordination behaviors. As the coordination behaviors of different teammates may vary significantly or even conflict with each other, it can be challenging to train a single policy network that coordinates effectively with all teammates. Additionally, when training with a newly generated teammate, the agent's policy may lose the ability to coordinate with previous ones due to network parameter updates, i.e., catastrophic forgetting.

To address these challenges, SEMDIV adopts a multi-head network architecture (Kessler et al., 2022; Yuan et al., 2024) and empowers the agents with continual learning ability. For each individual agent $\pi^{i \in \mathcal{N}_{\text{ag}}}$, the policy network is decomposed into a feature extractor $f_{\phi^i}$ and multiple policy heads $\{h_{\psi^{i,j}}\}_{j=1}^n$, where $n = |\{\boldsymbol{\pi}_1^{\text{tm}}, \ldots, \boldsymbol{\pi}_n^{\text{tm}}\}|$ represents the number of valid teammates generated up to the $n^{\text{th}}$ iteration. For simplicity, we ignore invalid teammates and

assume all teammates in $\mathcal{P}_n$ are valid ($I_m = \mathtt{True}$) in this part. For a new generated teammate $\boldsymbol{\pi}_{n+1}^{\mathrm{tm}}$ trained by reward $\hat{r}^{n+1}$ to demonstrate behavior $b_{n+1}$, SEMDIV first instantiates a new policy head $h_{\psi^i, n+1}$ for the agent's coordination with this new teammate. The joint agent policy $\boldsymbol{\pi}^{\mathrm{ag}} = \prod_{i \in \mathcal{N}_{\mathrm{ag}}} \pi^i = \prod_{i \in \mathcal{N}_{\mathrm{ag}}} f_{\phi^i} \circ h_{\psi^i, n+1}$ is then trained to coordinate with $\boldsymbol{\pi}_{n+1}^{\mathrm{tm}}$ by maximizing the objective:

$$J(\boldsymbol{\pi}^{\mathrm{ag}}, \boldsymbol{\pi}_{n+1}^{\mathrm{tm}}) = \mathbb{E}_{s_t, \boldsymbol{a}_t} \left[ \sum_t \gamma^t \left( r_t + \lambda_2 \hat{r}_t^{n+1} \right) \right], \quad (3)$$

where $\lambda_2$ is the same decaying factor with the one used in Equation (1). Different checkpoints of $\boldsymbol{\pi}_{n+1}^{\mathrm{tm}}$ are utilized for sampling to improve generalization. During training, the policy heads $\{h_{\psi^i, j}\}_{j=1}^n$ remain fixed, and gradients only propagate through $f_{\phi^i}$ and the new head $h_{\psi^i, n+1}$. Since the feature extractors $f_{\phi^i}$ are already well-trained to capture the common features of the task, $\boldsymbol{\pi}^{\mathrm{ag}}$ can quickly adapt to new teammates. However, $\boldsymbol{\pi}^{\mathrm{ag}}$ may lose the coordinate ability with previous teammates if $f_{\phi^i}$ updates dramatically, i.e., catastrophic forgetting. So, SEMDIV applies a regularization term to constrain the update, forming the final objective for training the joint agent policy:

$$\max_{\substack{\phi^i, \psi^{i, n+1} \\ i \in \mathcal{N}_{\mathrm{ag}}}} J(\boldsymbol{\pi}^{\mathrm{ag}}, \boldsymbol{\pi}_{n+1}^{\mathrm{tm}}) - \alpha \frac{1}{|\mathcal{N}_{\mathrm{ag}}|} \sum_{i \in \mathcal{N}_{\mathrm{ag}}} ||\phi^i - \bar{\phi}^i||_p, \quad (4)$$

where $J$ is the objective defined in Equation (3), $\alpha$ is a hyperparameter, $\bar{\phi}^i$ is the snapshot of parameters $\phi^i$ after training with the last teammate $\boldsymbol{\pi}_n^{\mathrm{tm}}$, and $|| \cdot ||_p$ represents the $l_p$ norm. This learning framework effectively balances the need to adapt to new teammates while preserving the ability to coordinate with previous ones. It has excellent scalability as the number of diverse teammates increases during training. Once the training process is complete, SEMDIV produces a joint agent policy $\boldsymbol{\pi}^{\mathrm{ag}}$ with a set of policy heads $\{h_{\psi^i, j}\}$, each tailored to coordinate with a class of teammates exhibiting a specific coordination behavior $b_j$. It is worth noting that, the agents are equipped with continual learning ability to adapt to future teammates that may appear after this training process, showcasing potential for online real-world applications.

During the execution phase, the agents need to coordinate with an unseen teammate $\boldsymbol{\pi}^{\mathrm{tm}}$ with coordination behavior $b$ described in natural language. SEMDIV utilizes an LLM to select the optimal policy head for the agents before rollout. This LLM selector takes the task description, learned behaviors $\{b_j \mid I_j = \mathtt{True}\}$, behavior $b$, and an instruction as prompts. The instruction is a sentence like "*select the policy that can best coordinate with the teammate*". Then, the LLM outputs the index $k$ of the selected head $h_{\psi^i, k}$. Finally, each individual agent $i$ uses $\pi^i = f_{\phi^i} \circ h_{\psi^i, k}$ to effectively coordinate with teammate $\boldsymbol{\pi}^{\mathrm{tm}}$. This approach enables the agents to adapt to the teammate through language-based

reasoning, avoiding the need for trial-and-error interactions and significantly improving efficiency. The prompts for this LLM are provided in Appendix F.5.

## 4. Experiments

In this section, we conduct a series of experiments to address the following questions: (1) Can SEMDIV effectively coordinate with unseen teammates who provide descriptions of their coordination behaviors (Section 4.2)? (2) How does SEMDIV operate in detail during a single run (Section 4.3)? (3) Can baselines achieve the performance of SEMDIV by increasing the population size (Section 4.4)?

### 4.1. Environments, Teammates, and Baselines

We evaluate SEMDIV and baseline methods across four classic multi-agent coordination environments. The first is **Level-Based Foraging (LBF)** (Papoudakis et al., 2021), a grid-world scenario where agents coordinate to collect food items together. Next, we introduce a modified version of the **Predator-Prey (PP)** (Lowe et al., 2017) environment, incorporating two prey types to enhance complexity. We then conduct experiments using the **StarCraft Multi-Agent Challenge-v2 (SMACv2)** (Ellis et al., 2023), which tasks agents with controlling StarCraft units to defeat enemies controlled by the game's built-in AI. SMACv2 improves upon SMAC (Samvelyan et al., 2019) by introducing features like randomized start positions, making it significantly more challenging. Finally, we test in **Google Research Football (GRF)** (Kurach et al., 2020), where agents control football players aiming to score through diverse tactics. Detailed introduction are provided in Appendix D.1.

In each environment, we train five teammates exhibiting distinct and representative coordination behaviors. For example, in GRF, we train teammates that prefer scoring after completing one or two passes. These teammates, along with their behavior descriptions, remain entirely unknown to the tested methods during training, ensuring an unbiased performance evaluation. To assess whether agents can effectively coordinate with these teammates to complete tasks, we measure the task success rates, denoted as R1. Additionally, we evaluate the success rate of agents in satisfying the teammates' preferred coordination behaviors, denoted as R2. Detailed introduction of the testing teammates are illustrated in Appendix D.2.

Next, we present the implementation details of SEMDIV and the baselines for comparison. In our experiments, we employ GPT-4o as the LLM[2]. For MARL algorithms, we utilize MAPPO (Yu et al., 2022) for GRF and VDN (Sunehag et al., 2018) for other environments. We first compare

---

[2]We use the `gpt-4o-2024-08-06` model via APIs at `https://platform.openai.com/docs/guides/gpt`.

*Table 1.* Coordination performance (mean $\pm$ std) with unseen teammates across four environments. "R1" and "R2" represent the success rates of task completion and agents satisfying the teammates preferred coordination behaviors, respectively. The best result in each column, excluding performance upper bounds of SEMDIV (denoted in gray), is highlighted in **bold**.

| Methods | LBF | | PP | | SMACv2 | | GRF | | Average | |
|---|---|---|---|---|---|---|---|---|---|---|
| | R1 | R2 | R1 | R2 | R1 | R2 | R1 | R2 | R1 | R2 |
| Oracle | 1.00 | 1.00 | 0.91 | 0.90 | 0.94 | 0.93 | 0.95 | 0.95 | 0.95 | 0.95 |
| SEMDIV | **0.90** $_{\pm 0.05}$ | **0.90** $_{\pm 0.05}$ | **0.72** $_{\pm 0.03}$ | **0.54** $_{\pm 0.10}$ | **0.65** $_{\pm 0.02}$ | **0.64** $_{\pm 0.02}$ | **0.67** $_{\pm 0.08}$ | **0.62** $_{\pm 0.07}$ | **0.74** | **0.68** |
| SEMDIV-Dist | 0.45 $_{\pm 0.14}$ | 0.45 $_{\pm 0.14}$ | 0.51 $_{\pm 0.03}$ | 0.28 $_{\pm 0.05}$ | 0.24 $_{\pm 0.08}$ | 0.23 $_{\pm 0.08}$ | 0.47 $_{\pm 0.20}$ | 0.37 $_{\pm 0.16}$ | 0.42 | 0.33 |
| SEMDIV-R1 | 0.91 $_{\pm 0.04}$ | 0.91 $_{\pm 0.04}$ | 0.76 $_{\pm 0.01}$ | 0.53 $_{\pm 0.04}$ | 0.70 $_{\pm 0.00}$ | 0.69 $_{\pm 0.01}$ | 0.88 $_{\pm 0.06}$ | 0.62 $_{\pm 0.08}$ | 0.81 | 0.69 |
| SEMDIV-R2 | 0.91 $_{\pm 0.04}$ | 0.91 $_{\pm 0.04}$ | 0.74 $_{\pm 0.01}$ | 0.58 $_{\pm 0.06}$ | 0.70 $_{\pm 0.00}$ | 0.69 $_{\pm 0.01}$ | 0.78 $_{\pm 0.08}$ | 0.73 $_{\pm 0.05}$ | 0.78 | 0.73 |
| Macop-R1 | 0.82 $_{\pm 0.10}$ | 0.81 $_{\pm 0.11}$ | 0.58 $_{\pm 0.02}$ | 0.23 $_{\pm 0.00}$ | 0.48 $_{\pm 0.03}$ | 0.45 $_{\pm 0.03}$ | 0.59 $_{\pm 0.15}$ | 0.44 $_{\pm 0.04}$ | 0.62 | 0.48 |
| Macop-R2 | 0.82 $_{\pm 0.10}$ | 0.81 $_{\pm 0.11}$ | 0.54 $_{\pm 0.01}$ | 0.25 $_{\pm 0.00}$ | 0.47 $_{\pm 0.03}$ | 0.45 $_{\pm 0.03}$ | 0.56 $_{\pm 0.15}$ | 0.45 $_{\pm 0.03}$ | 0.60 | 0.49 |
| SEMDIV-PBT | 0.64 $_{\pm 0.02}$ | 0.64 $_{\pm 0.02}$ | 0.70 $_{\pm 0.01}$ | 0.31 $_{\pm 0.01}$ | 0.61 $_{\pm 0.01}$ | 0.61 $_{\pm 0.01}$ | 0.57 $_{\pm 0.30}$ | 0.39 $_{\pm 0.12}$ | 0.63 | 0.49 |
| Macop-PBT | 0.61 $_{\pm 0.00}$ | 0.60 $_{\pm 0.02}$ | **0.72** $_{\pm 0.03}$ | 0.33 $_{\pm 0.03}$ | 0.56 $_{\pm 0.04}$ | 0.54 $_{\pm 0.03}$ | 0.49 $_{\pm 0.24}$ | 0.35 $_{\pm 0.10}$ | 0.60 | 0.46 |
| FCP | 0.46 $_{\pm 0.22}$ | 0.43 $_{\pm 0.20}$ | 0.57 $_{\pm 0.23}$ | 0.21 $_{\pm 0.15}$ | 0.40 $_{\pm 0.05}$ | 0.37 $_{\pm 0.06}$ | 0.50 $_{\pm 0.25}$ | 0.36 $_{\pm 0.12}$ | 0.48 | 0.34 |
| MEP | 0.57 $_{\pm 0.08}$ | 0.56 $_{\pm 0.08}$ | 0.70 $_{\pm 0.01}$ | 0.31 $_{\pm 0.01}$ | 0.55 $_{\pm 0.04}$ | 0.47 $_{\pm 0.02}$ | 0.50 $_{\pm 0.26}$ | 0.35 $_{\pm 0.14}$ | 0.58 | 0.42 |
| LIPO | 0.54 $_{\pm 0.00}$ | 0.51 $_{\pm 0.02}$ | 0.69 $_{\pm 0.02}$ | 0.31 $_{\pm 0.01}$ | 0.45 $_{\pm 0.10}$ | 0.38 $_{\pm 0.06}$ | 0.51 $_{\pm 0.25}$ | 0.37 $_{\pm 0.12}$ | 0.55 | 0.39 |
| LLM-Agent | 0.88 $_{\pm 0.05}$ | 0.88 $_{\pm 0.05}$ | 0.71 $_{\pm 0.09}$ | 0.53 $_{\pm 0.08}$ | 0.35 $_{\pm 0.10}$ | 0.35 $_{\pm 0.10}$ | 0.14 $_{\pm 0.09}$ | 0.12 $_{\pm 0.09}$ | 0.52 | 0.47 |

SEMDIV with classic two-stage population-based training (PBT) methods that induce diversity at the policy level, including FCP (Strouse et al., 2021), MEP (Zhao et al., 2023), and LIPO (Charakorn et al., 2023). These methods train a population of diverse teammates using different techniques in the first stage, and use them to train agents in the second stage. Then, we compare SEMDIV with Macop (Yuan et al., 2023a), which employs an iterative process similar to SEMDIV but generates new teammates by minimizing compatibility with agents. For a fair comparison, we derive a total of 6 teammates and extract their three checkpoints: the initial, middle, and final stages of training (Strouse et al., 2021). This results in 3 checkpoints per teammate and a total of 18 teammate policies for agent training across all methods. To analyze the quality of the generated teammates and the impact of the multi-head architecture, we use the teammates of SEMDIV and Macop as the first-stage teammates in PBT methods, denoted as {SEMDIV, Macop}-PBT. To investigate the head selection module, we include {SEMDIV, Macop}-R1 and -R2, which report the results of the heads with the highest R1 or R2 values, serving as upper bounds. Additionally, we introduce SEMDIV-Dist, an ablation of SEMDIV that selects heads based on the distance between embeddings of behavior descriptions, computed using a T5-XL model (Chung et al., 2024). Since SEMDIV combines the strengths of MARL and LLMs, we also include a baseline LLM-Agent that uses LLM only, to assess the necessity of MARL. All methods are evaluated over three random seeds. Finally, we report the self-play performance of testing teammates as upper bounds (Oracle). Further details for SEMDIV and the baselines are in Appendix B and C.

## 4.2. Competitive Results

In this section, we present the overall results of SEMDIV, its ablations, and the baseline methods when coordinating with unseen teammates across four environments. As shown in Table 1, the classic method FCP demonstrates poor performance, due to its limited ability to generate sufficiently diverse teammates. In contrast, methods that incorporate additional diversity objectives, such as MEP and LIPO, show improved performance, highlighting the importance of fostering distinct coordination behaviors that cannot be captured by simply training with varied seeds. However, all these two-stage PBT methods exhibit limited coordination ability. When we replace the first-stage teammates with those generated by SEMDIV or Macop (*-PBT), performance improves significantly, suggesting that the two-stage framework struggles to generate sufficiently diverse teammates without considering the agents. Among these PBT methods, SEMDIV-PBT achieves the best results (see the third block of the table), demonstrating that SEMDIV generates teammates with superior quality and diversity.

Further analysis reveals that a single policy network is insufficient to effectively adapt to all distinct teammates, i.e., the multi-modality issue. The multi-head versions of SEMDIV and Macop (second table block) outperform their PBT counterparts, indicating that multi-head architecture can address this issue. Next, SEMDIV consistently outperforms all baselines, demonstrating the effectiveness of its semantically diverse teammate generation. In the multi-head settings, SEMDIV leverages an LLM to understand the behaviors and coordination tasks, thus selecting matched policy heads.

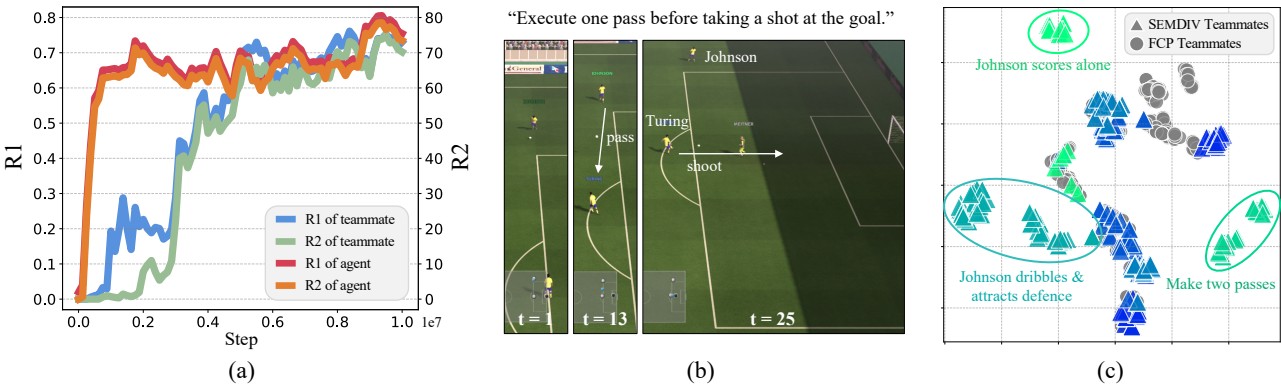

*Figure 3.* A case study in the GRF environment. (a) Learning curves of the teammate and the agent in the first iteration of SEMDIV. (b) An episode where the first generated teammate successfully scores a goal and demonstrates the desired coordination behavior. (c) Trajectories visualization of the 12 teammates generated by SEMDIV and FCP.

It achieves results comparable to the upper bounds of -R1 and -R2, and outperforms the best baseline Macop by 19% for R1 and 39% for R2. In contrast, SEMDIV-Dist selects heads based on embedding distances between behavior descriptions, and shows significant performance degradation, indicating that language embedding similarity alone is insufficient to address the complex task of head selection. Although SEMDIV still falls short of the Oracle baseline, we can bridge the gap by generating more teammates or incorporating additional diversity objectives.

Additionally, while LLM-Agent performs comparably to SEMDIV in simpler tasks such as LBF and PP, it experiences a severe performance degradation in more complex environments, highlighting the necessity of incorporating task-specific reinforcement learning for successful multi-agent coordination. More experimental results, including performance of each testing teammate, are in Appendix E.

### 4.3. Case Study

To illustrate the functionality of SEMDIV in detail, we present a case study that demonstrates the teammate generation process, agent training, and evaluation with an unseen teammate during a single run in the GRF environment.

At the beginning, the LLM behavior generator takes the designed prompt as input, and outputs a possible coordination behavior: *execute one pass before taking a shot at the goal*. Based on this behavior and the context of the football game, the LLM reward generator outputs the corresponding reward function in Python, as shown in the example in Figure 4.

The generated function correctly utilizes the provided environment attributes to encourage the teammate to learn the specified passing tactic. The inclusion of well-documented comments enhances the reward's interpretability. This func-

```python
def reward1(self) -> float:
    # Check if the score event happens at this step
    if self.score:
        # Check if there is exactly one pass in the history
        if len(self.pass_history) == 1:
            # Check if the pass is between the two players
            # ... (Codes omitted for clarity)
            # Large reward to reinforce the desired behavior
            return 100.0
    # Default return, no extra reward in other cases
    return 0.0
```

*Figure 4.* Python code example for reward calculation.

tion is then incorporated into the reward wrapper class. Subsequently, SEMDIV applies the MAPPO (Yu et al., 2022) algorithm to train the teammates to maximize both the task reward and the generated reward, as defined in Equation (1). The training results are shown in Figure 3(a). Upon completing training, SEMDIV verifies the validity of the learned teammate policy. First, as shown in the learning curves, at the early stage of training, the teammate occasionally scores goals without completing the desired passing behavior, leading to a discrepancy between the blue and green curves. As training goes, the teammate successfully learns to score while maximizing the generated reward. Second, trajectory data is extracted and translated into natural language, producing a summary: *"In this episode, Johnson passed to Turing, and finally successfully scored a goal. The player who scored the goal is Turing ⋯ "* Based on this summary, an LLM confirms that the policy aligns with the intended coordination behavior. Key steps of this episode are visualized in Figure 3(b). Third, the similarity check is skipped as this is the first teammate. This coordination behavior and its corresponding teammate policy are thus validated as suitable for training the agent.

Next, SEMDIV creates a new policy head for the agent, and trains it to coordinate with this teammate, as defined in

Equation (4). For this initial teammate, the regularization coefficient $\alpha$ is set to 0. The agent efficiently learns to score goals with the teammate while executing the intended passing tactic, resulting in rapidly rising and overlapping learning curves shown in red and orange. This process is repeated iteratively until the agent is trained with six distinct valid teammates.

To assess the impact of the semantic-level exploration technique on enhancing diversity among teammate policies, we visualize the generated trajectories. Specifically, we collect 100 trajectories for each of the six valid teammates, totaling 600 trajectories. For comparison, we also gather an equivalent dataset from six teammates generated during a run using FCP (Strouse et al., 2021). From these trajectories, we extract those that result in a goal, convert them into vector representations, and apply t-SNE (Van der Maaten & Hinton, 2008) for visualization. As shown in Figure 3(c), the projection of SEMDIV exhibits a broader and more dispersed coverage compared to FCP (highlighted in circles). This confirms that semantic-level exploration significantly enhances the coverage of the teammate policy space, ultimately enhancing the agent's coordination.

Finally, the agent is evaluated with an unseen teammate. For example, a teammate joins the team as Turing, the player at the center. Our agent controls the other player, Johnson, and needs to coordinate with Turing. Before the game begins, Turing describes his/her desired coordination behavior: *"I prefer to score myself."* The LLM head selector takes the task description, Turing's desired behavior, and behaviors the agent have learned, as inputs. It inferences that *"This policy (the one described above) fulfills Turing's desire to score, as it allows him to set up for a shot after receiving a pass."*, and selects the optimal head. Equipped with the selected head, the team achieves an 88% scoring rate with the teammate, with all goals scored by Turing. This case study highlights the effectiveness of SEMDIV in generating diverse teammate policies, enabling efficient coordination even with unseen teammates.

### 4.4. The Impact of the Number of Teammates

One of the key factors affecting performance is the number of teammates with whom the agents train. To investigate its impact, we run SEMDIV, its variant SEMDIV-PBT, and the baseline FCP with different numbers of training teammates, and assess the agents' performance with the testing teammates. As shown in Figure 5, when training with only one teammate, these methods degenerate to the same setting, showing almost identical performance. As the number of teammates increases, SEMDIV-PBT outperforms FCP with the same number of training teammates, achieving comparable or even superior results to FCP with a significantly larger number of 48 teammates. This demonstrates that

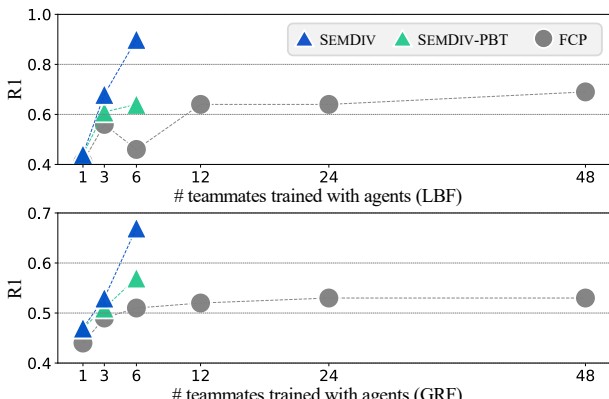

*Figure 5.* Coordination performance with testing teammates when agents train with various numbers of generated teammates.

generating semantically diverse teammates not only enables more efficient exploration of the teammate policy space but also facilitates the discovery of coordination behaviors that policy-level exploration alone cannot cover. For instance, in the GRF environment, we observe that FCP and other baselines fail to discover complex tactics that pass multiple times. Furthermore, with its multi-head architecture, SEMDIV scales more effectively with the number of teammates, achieving significantly better performance than SEMDIV-PBT. This highlights the importance of a specialized design that allows for rapid adaptation to unseen teammates.

## 5. Related Work

In open multi-agent environments, the important factors of the environment or the multi-agent system may change unexpectedly (Yuan et al., 2023b). To handle the change of teammates, recent research in areas such as ad-hoc teamwork (Mirsky et al., 2022; Wang et al., 2024a) and zero-shot coordination (Treutlein et al., 2021) has emerged. This line of work includes training paradigm design (Hu et al., 2020; Strouse et al., 2021), diverse teammate generation (Lupu et al., 2021; Zhao et al., 2023; Charakorn et al., 2023; Yuan et al., 2023a; Rahman et al., 2023; 2024; Cui et al., 2023; Sarkar et al., 2023), investigation of human bias (Yu et al., 2023a; Hu & Sadigh, 2023), goal deduction (Zhang et al., 2024d), and policy co-evolution for heterogeneous settings (Xue et al., 2024). Researchers also develop benchmarks (Wang et al., 2024b) to evaluate these methods. This paper further delves into this line of work utilizing the power of LLMs to enhance teammates' semantic diversity.

LLMs have recently gained significant attention in multi-agent tasks due to their advanced capabilities in natural language processing and planning (Guo et al., 2024). One line of work utilize LLMs for language agents communication (Park et al., 2023; Guan et al., 2024; Zhang et al., 2024b; Li et al., 2023a; Du et al., 2024; Wang et al., 2024c).

Some other works utilize LLMs as multi-agent task planners, which can be classified into several key areas, including MARL subgoal generation (Li et al., 2023b), multi-agent path finding (Chen et al., 2024a), and multi-robot task planning (Liu et al., 2024b; Chang et al., 2024). Despite these advancements, LLMs still face challenges in handling low-level coordination in multi-agent settings. Rather than directly deploying LLMs as coordinating agents, we leverage their capabilities to generate diverse teammates and adapting policies, thereby combining the strengths of LLMs with MARL. We discuss more related work in Appendix A.

## 6. Final Remarks

We propose a novel framework of LLM-assisted Semantically Diverse Teammate Generation (SEMDIV) for efficient multi-agent coordination. The framework utilizes LLMs to discover diverse coordination behaviors described in natural language, facilitating the training of teammate policies aligning with these behaviors. Agents train with these teammates in a continual learning process, developing policies tailored to the coordination behaviors and enabling rapid adaptation to testing teammates. Empirical results across various environments and with unseen teammates provide strong evidence of SEMDIV's effectiveness. Looking ahead, as more advanced MARL techniques and LLMs emerge with enhanced performance, SEMDIV has the potential to further improve agent generalization in complex real-world coordination scenarios, such as embodied multi-agent tasks (Feng et al., 2025) for real-world applications.

## Acknowledgements

This work was supported by the NSFC (62495093, U24A20324) and Jiangsu Science Foundation (BK20241199, BK20243039). We thank Tencent AI Arena for their support, and the anonymous reviewers for their support on improving the paper.

## Impact Statement

The goal of the work presented in this paper is to advance the development of cooperative multi-agent reinforcement learning. The proposed framework is intended to enhance the generalization of coordinating agents, providing an effective approach for future research on open multi-agent systems. Furthermore, the work presented does not raise any additional ethical concerns, and thus no special discussion on ethical issues is required.

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

# A. More Related Work

**Cooperative Multi-Agent Reinforcement Learning (MARL)**   Many real-world problems, particularly those that are large-scale and complex, are inherently suited to be modeled as multi-agent systems (MASs) rather than single-agent systems due to their efficiency and practicality in addressing intricate challenges (Dorri et al., 2018). Multi-agent reinforcement learning (MARL) (Zhang et al., 2021) has emerged as a powerful framework for tackling these problems, leveraging the problem-solving capabilities of deep reinforcement learning (Wang et al., 2022). When agents within a MAS share common objectives, the problem falls under the category of cooperative MARL (Oroojlooy & Hajinezhad, 2023), which has demonstrated significant success across diverse domains such as autonomous driving (Zhang et al., 2024c), domain calibration (Jiang et al., 2024), and financial trading (Huang et al., 2024). Recent advancements in MARL have introduced a variety of approaches to improve agent coordination. These include policy-based methods such as MADDPG (Lowe et al., 2017) and MAPPO (Yu et al., 2022), value-based techniques like VDN (Sunehag et al., 2018) and QMIX (Rashid et al., 2018), as well as innovative approaches leveraging architectures such as the transformer (Wen et al., 2022). These methods have demonstrated exceptional coordination capabilities in diverse tasks, including SMAC (Samvelyan et al., 2019) and GRF (Kurach et al., 2020). In this paper, our method focuses on enhancing the generalization abilities of coordinating agents, aiming to improve their adaptability and performance across a wider range of potential teammates.

**Large Language Models (LLMs) for RL**   The integration of large language models (LLMs) into reinforcement learning (RL) has emerged as a promising research direction (Cao et al., 2024), leveraging the rich semantic understanding and generalization capabilities of LLMs to enhance decision-making processes. Recent studies have explored the use of LLMs for tasks such as processing and translating task information (Paischer et al., 2022; Choi et al., 2023; Pang et al., 2023; Spiegel et al., 2024), to reduce the burden of network updates. Another line of work utilizes LLMs as reward generator (Carta et al., 2022; Kwon et al., 2023; Wu et al., 2023; Yu et al., 2023b; Du et al., 2023) to guide RL algorithms. Specifically, some approaches (Xie et al., 2024; Ma et al., 2024a;b) explicitly generate executable codes as reward functions. LLMs are also utilized as world models (Pang et al., 2024; Chen et al., 2024b; Lin et al., 2024; Zhang et al., 2024a) as they are trained with rich real-world context, enhancing the sample efficiency of RL. In our work, we mainly utilize LLMs to propose coordination behaviors described in natural language, reward generation, and behavior-trajectory alignment verification.

# B. Implementation Details of SEMDIV

In this section, we present the implementation details of SEMDIV. The `gpt-4o-2024-08-06` model is utilized as the LLM. For MARL algorithms, we employ VDN (Sunehag et al., 2018) for the LBF, PP, and SMACv2 environments, and MAPPO (Yu et al., 2022) for GRF. Specifically, our VDN implementation is based on the PyMARL codebase (Samvelyan et al., 2019)[3]. We adopt parameter sharing in the agent network architecture. The feature extractor $f_\phi^i$ is designed as a 3-layer MLP followed by a GRU (Cho et al., 2014), while the policy head $h_{\psi^i}$ is a 3-layer MLP. Both the MLP and GRU have a hidden dimension of 64. The policy head processes the feature extractor's output to generate Q-values for all actions, which are subsequently aggregated by summing individual agents' Q-values to compute the joint Q-value. The architecture for teammate networks mirrors this design, differing only in having a single policy head. For MAPPO, we build upon the HARL codebase (Liu et al., 2024a)[4]. Unlike VDN, parameter sharing is not applied by default settings. For the actor networks, the final two-layer MLP serves as the policy head, and the remaining components form the feature extractor. The critic networks are left unmodified. A single run of SEMDIV incurs a cost of approximately $0.10 for OpenAI APIs and $300 for the full project.

We use the default hyperparameter settings of PyMARL and HARL, e.g., the learning rates of the algorithms. The selection of the special hyperparameters introduced in this paper, e.g., the training steps for each teammate, is listed in Table 2.

# C. Implementation Details of Baselines

We first compare SEMDIV with classic two-stage population-based training (PBT) methods, which train a population of teammates using different techniques in the first stage, and use them to train agents in the second stage. **FCP** (Strouse et al., 2021) first trains a population of teammate policies using different random seeds independently. Then, it trains the agents by pairing them with three checkpoints of each teammate: the initial, middle, and final stages of training. In our implementation,

---

[3] https://github.com/oxwhirl/pymarl
[4] https://github.com/PKU-MARL/HARL

*Table 2.* Hyperparameters in the experiments.

| Hyperparameter | Value |
|---|---|
| Training steps for one teammate | $10^5$ (LBF), $5 \times 10^5$ (PP), $10^6$ (SMACv2), $10^7$ (GRF) |
| Number of teammates trained with agents | 6 |
| Training steps for agents with one teammate | $3 \times 10^5$ (LBF), $5 \times 10^5$ (PP), $10^6$ (SMACv2), $10^7$ (GRF) |
| Threshold for teammate performance verification | 0.3 (LBF, PP), 0.5 (SMACv2, GRF) |
| Maximum attempts for generating a teammate policy | 2 |
| Threshold $\epsilon$ for teammate novelty verification in Equation (2) | 0.2 |
| Coefficient $\alpha$ for regularizing feature extractors in Equation (4) | 500 |

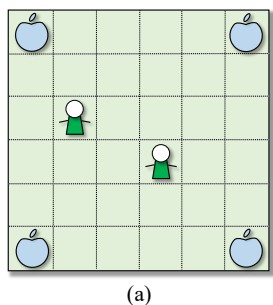 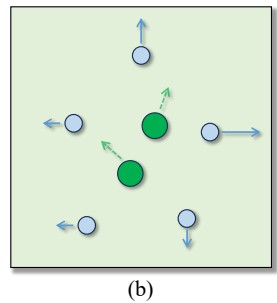 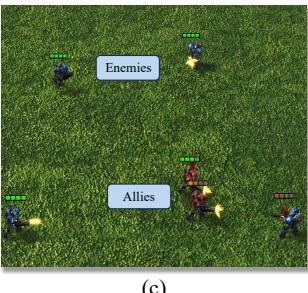 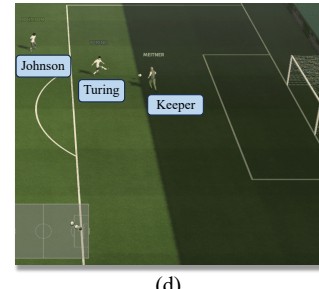

| (a) | (b) | (c) | (d) |

*Figure 6.* Environments used in this paper. (a) Level-based Foraging (LBF) (Papoudakis et al., 2021). (b) Predator-Prey (PP) (Lowe et al., 2017). (c) StarCraft Multi-Agent Challenge-v2 (SMACv2) (Ellis et al., 2023). (d) Google Research Football (GRF) (Kurach et al., 2020).

we set the population size as 6, and train the teammates and agents until convergence. Based on FCP, **MEP** (Zhao et al., 2023) applies a entropy term $\mathcal{H}(\bar{\pi}(\cdot \mid s_t))$ when training the population, where $\bar{\pi}(\boldsymbol{a}_t \mid s_t) = \frac{1}{n}\sum_{i=1}^{n}\pi^i(a_t \mid s_t)$. **LIPO** (Charakorn et al., 2023) replaces this term with $J_{\text{LIPO}} = -\sum_{i \neq j} J(\boldsymbol{\pi}_{\text{tm}}^i, \boldsymbol{\pi}_{\text{tm}}^j)$. We set the weights for these optimization terms as 0.001 across all environments. The rest implementation of MEP and LIPO remains the same as FCP. During execution, these methods directly deploy the only agent policy for coordination, without explicit adaptation process. While two-stage methods generate teammates before training agents, another baseline **Macop** (Yuan et al., 2023a) adopts an agents-centric paradigm, where it alternatively generates new teammates and trains the multi-head agents, inducing diversity by reducing the compatibility $J(\boldsymbol{\pi}_{\text{ag}}, \boldsymbol{\pi}_{\text{tm}})$ between the teammates and the current agents. To select the policy heads for execution, Macop must collect multiple episodes to gather adequate information, hindering its deployment in costly tasks. In our experiments, we report Macop's results of the heads that maximize the R1 or R2 values. Since SEMDIV combines the strengths of MARL and LLMs, we also include a baseline LLM-Agent that directly uses an LLM as the policy, to assess the necessity of MARL. The prompts for this LLM are provided in Appendix F.6.

## D. Experiment Details

In this section, we provide more details about the experiments, including the environments and the unseen testing teammates.

### D.1. Environments

We use four classic cooperative MARL environments with diverse coordination behaviors, as shown in Figure 6.

**Level-based Foraging (LBF)** (Papoudakis et al., 2021) is a discrete game where agents of varying levels navigate a grid to collect foods with corresponding levels. Each agent moves one cell at a time in one of the four cardinal directions: {up, left, down, right}. Agents are rewarded with 1 when they are positioned one cell away from a food item and the sum of their levels matches or exceeds the food's level. In this work, we use a $6 \times 6$ grid-world setup with four level-2 foods located at $(0,0)$, $(0,5)$, $(5,0)$, and $(5,5)$. Two level-1 agents are randomly spawned at cells $\{(2,2),(2,3),(3,2),(3,3)\}$. An episode terminates when agents collect one food or after nine steps. Coordination is essential as agents must observe their teammate's preferences and collaborate to collect the foods.

**Predator-Prey (PP)** is a widely-used benchmark from the Multiagent Particle Environment (MPE) (Lowe et al., 2017), where predators and prey are represented as circles on a 2D plane. Agents controlling predators can accelerate in one of four

directions {up, left, down, right} to pursue prey, which employ a heuristic policy to evade the nearest predator. We extend this benchmark to include five prey: two *stags*, which require both predators to simultaneously capture them, and three *rabbits*, which can be captured by a single predator. Capturing a stag rewards the agents with 1, while capturing a rabbit rewards 0.5. However, if only one predator attempts to capture a stag, the team is penalized with a reward of -0.01. An episode terminates when predators catch one stag or two rabbits, or after twenty steps. Effective coordination is required for agents to adapt to their teammate's strategies and successfully hunt the prey.

**StarCraft Multi-Agent Challenge-v2 (SMACv2)** (Ellis et al., 2023) is an extended version of SMAC (Samvelyan et al., 2019). In this environment, ally units (agents) must defeat enemy units controlled by the game's built-in AI. Agents receive positive rewards for dealing damage, eliminating enemies, and winning battles, while incurring negative rewards for receiving damage, losing units, or being defeated. SMACv2 introduces randomized start positions, increasing the difficulty and variability of scenarios. In our experiments, two ally marine units face four enemy marine units. The predefined `surrounded` start position requires agents to move cohesively, focus fire on individual enemies, and adapt to their teammate's combat strategies to win battles. Agents are deemed successful if they eliminate at least one enemy.

**Google Research Football (GRF)** (Kurach et al., 2020) is a physics-based 3D football simulator that closely replicates the rules and dynamics of real-world football. Agents can perform actions such as passing, defending, and shooting. We design a scenario where two agents control players, Johnson and Turing, attempting to score from the edge of the penalty box. Johnson starts with the ball on the wing, while Turing positions centrally, facing the goalkeeper (Meitner). The team receives a reward of 1 for scoring a goal and small rewards for getting closer to the goal. An episode terminates when a goal is scored, the goalkeeper gains possession of the ball, the ball goes out of bounds, or after 100 steps. Effective collaboration between Johnson and Turing is required to win the game.

### D.2. Testing Teammates

We evaluate the generalization capabilities of different methods by training five manually designed teammates with distinct and representative coordination behaviors for each environment. These teammates, along with their behavior descriptions, remain entirely unknown to the tested methods during training, ensuring an unbiased performance evaluation. In LBF, we train four teammates that specialize in collecting one specific food, with descriptions such as *"I prefer to collect food A/B/C/D"*, and one teammate that *"collects the food closest to our average position"*. In PP, we train five teammates with preferences for capturing specific prey, described as *"I prefer to catch {prey}"*. These include teammates that prioritize stag 1, stag 2, rabbit 1&2, rabbit 1&3, and rabbit 2&3. In SMACv2, we design five teammates similar to those in LBF, where the "foods" are replaced by "enemies". In GRF, we train teammates with behaviors such as letting Johnson or Turing score, or scoring after passing 0, 1, or 2 times. Given the slight heterogeneity between the two players, we evaluate scenarios where the teammate controls either Turing or Johnson and report the average results. The pronoun in the description changes depending on the player controlled by the teammate. For example, when the teammate trained to let Turing score controls Turing, the description is *I prefer to score myself,"* instead of *I prefer to let Turing score."*. This imposes a higher requirement on the methods' ability to understand teammate behaviors.

## E. More Experiment Results

We present more experiment results, including the coordination performance with each unseen teammate, the impact of ambiguity in the natural language descriptions of teammates' behaviors, and the impact of the quality of the LLMs.

### E.1. Results with Each Testing Teammate

In Section 4.2, we evaluate the agents from different methods in each environment by testing them with five unseen teammates and computing the average R1 and R2 values. To minimize randomness, we repeat the experiments across three random seeds and report the mean ± standard deviation of the average R1 and R2 values, as summarized in Table 1. Here, we present the performance with each individual testing teammate in Table 3–7. As shown in these tables, certain baselines exhibit unstable performance, coordinating effectively with some teammates but failing with others. For instance, FCP achieves R1 and R2 values exceeding 0.7 with Teammates 1 and 5, yet falls below 0.5 with the remaining teammates. This suggests that the baselines struggle to capture specific coordination behaviors, leading to agents that overfit to teammates with limited diversity. In contrast, SEMDIV consistently delivers the best overall results across all methods, demonstrating its robustness and ability to adapt effectively to diverse teammates.

*Table 3.* Coordination performance (mean ± std) with five unseen teammates in the LBF environment. "R1" and "R2" represent the success rates of task completion and agents satisfying the teammates preferred coordination behaviors, respectively. The best result in each column, excluding performance upper bounds of SEMDIV (denoted in gray), is highlighted in **bold**.

| Methods | Teammate 1 | | Teammate 2 | | Teammate 3 | | Teammate 4 | | Teammate 5 | | Average | |
|---|---|---|---|---|---|---|---|---|---|---|---|---|
| | R1 | R2 | R1 | R2 | R1 | R2 | R1 | R2 | R1 | R2 | R1 | R2 |
| Oracle | 1.00 | 1.00 | 1.00 | 1.00 | 1.00 | 1.00 | 1.00 | 1.00 | 1.00 | 1.00 | 1.00 | 1.00 |
| SEMDIV | $0.91_{\pm0.07}$ | $0.91_{\pm0.07}$ | $\mathbf{0.90}_{\pm0.03}$ | $\mathbf{0.90}_{\pm0.03}$ | $\mathbf{0.95}_{\pm0.04}$ | $\mathbf{0.95}_{\pm0.04}$ | $0.87_{\pm0.02}$ | $0.87_{\pm0.02}$ | $0.89_{\pm0.15}$ | $0.89_{\pm0.15}$ | $\mathbf{0.90}$ | $\mathbf{0.90}$ |
| SEMDIV-t5 | $0.29_{\pm0.41}$ | $0.29_{\pm0.41}$ | $0.07_{\pm0.09}$ | $0.07_{\pm0.09}$ | $0.52_{\pm0.39}$ | $0.52_{\pm0.39}$ | $0.45_{\pm0.35}$ | $0.45_{\pm0.35}$ | $0.89_{\pm0.15}$ | $0.89_{\pm0.15}$ | 0.45 | 0.45 |
| SEMDIV-R1 | $0.92_{\pm0.07}$ | $0.92_{\pm0.07}$ | $0.93_{\pm0.06}$ | $0.93_{\pm0.06}$ | $0.95_{\pm0.04}$ | $0.95_{\pm0.04}$ | $0.87_{\pm0.02}$ | $0.87_{\pm0.02}$ | $0.94_{\pm0.08}$ | $0.93_{\pm0.09}$ | 0.91 | 0.91 |
| SEMDIV-R2 | $0.92_{\pm0.07}$ | $0.92_{\pm0.07}$ | $0.93_{\pm0.06}$ | $0.93_{\pm0.06}$ | $0.95_{\pm0.04}$ | $0.95_{\pm0.04}$ | $0.87_{\pm0.02}$ | $0.87_{\pm0.02}$ | $0.94_{\pm0.08}$ | $0.93_{\pm0.09}$ | 0.91 | 0.91 |
| Macop-R1 | $\mathbf{0.97}_{\pm0.02}$ | $\mathbf{0.97}_{\pm0.02}$ | $0.81_{\pm0.09}$ | $0.81_{\pm0.09}$ | $0.75_{\pm0.31}$ | $0.75_{\pm0.31}$ | $\mathbf{0.87}_{\pm0.12}$ | $\mathbf{0.87}_{\pm0.12}$ | $0.70_{\pm0.12}$ | $0.65_{\pm0.15}$ | 0.82 | 0.81 |
| Macop-R2 | $\mathbf{0.97}_{\pm0.02}$ | $\mathbf{0.97}_{\pm0.02}$ | $0.81_{\pm0.09}$ | $0.81_{\pm0.09}$ | $0.75_{\pm0.31}$ | $0.75_{\pm0.31}$ | $\mathbf{0.87}_{\pm0.12}$ | $\mathbf{0.87}_{\pm0.12}$ | $0.70_{\pm0.12}$ | $0.65_{\pm0.15}$ | 0.82 | 0.81 |
| SEMDIV-PBT | $0.69_{\pm0.25}$ | $0.69_{\pm0.25}$ | $0.51_{\pm0.13}$ | $0.51_{\pm0.13}$ | $0.33_{\pm0.07}$ | $0.33_{\pm0.07}$ | $0.65_{\pm0.26}$ | $0.65_{\pm0.26}$ | $\mathbf{1.00}_{\pm0.00}$ | $\mathbf{1.00}_{\pm0.00}$ | 0.64 | 0.64 |
| Macop-PBT | $0.68_{\pm0.22}$ | $0.68_{\pm0.22}$ | $0.51_{\pm0.08}$ | $0.51_{\pm0.08}$ | $0.47_{\pm0.15}$ | $0.47_{\pm0.15}$ | $0.57_{\pm0.22}$ | $0.57_{\pm0.22}$ | $0.83_{\pm0.08}$ | $0.77_{\pm0.14}$ | 0.61 | 0.60 |
| FCP | $0.74_{\pm0.28}$ | $0.74_{\pm0.28}$ | $0.46_{\pm0.30}$ | $0.46_{\pm0.30}$ | $0.37_{\pm0.26}$ | $0.37_{\pm0.26}$ | $0.47_{\pm0.31}$ | $0.47_{\pm0.31}$ | $0.79_{\pm0.22}$ | $0.70_{\pm0.22}$ | 0.57 | 0.55 |
| MEP | $0.57_{\pm0.40}$ | $0.57_{\pm0.40}$ | $0.27_{\pm0.38}$ | $0.27_{\pm0.38}$ | $0.75_{\pm0.17}$ | $0.75_{\pm0.17}$ | $0.41_{\pm0.18}$ | $0.41_{\pm0.18}$ | $0.85_{\pm0.15}$ | $0.79_{\pm0.15}$ | 0.57 | 0.56 |
| LIPO | $0.55_{\pm0.31}$ | $0.55_{\pm0.31}$ | $0.43_{\pm0.34}$ | $0.43_{\pm0.34}$ | $0.49_{\pm0.27}$ | $0.49_{\pm0.27}$ | $0.47_{\pm0.33}$ | $0.47_{\pm0.33}$ | $0.75_{\pm0.14}$ | $0.63_{\pm0.14}$ | 0.54 | 0.51 |
| LLM-Agent | $0.83_{\pm0.10}$ | $0.83_{\pm0.10}$ | $0.82_{\pm0.02}$ | $0.82_{\pm0.02}$ | $0.88_{\pm0.06}$ | $0.88_{\pm0.06}$ | $0.92_{\pm0.08}$ | $0.92_{\pm0.08}$ | $0.95_{\pm0.04}$ | $0.95_{\pm0.04}$ | 0.88 | 0.88 |

*Table 4.* Coordination performance (mean ± std) with five unseen teammates in the PP environment. "R1" and "R2" represent the success rates of task completion and agents satisfying the teammates preferred coordination behaviors, respectively. The best result in each column, excluding performance upper bounds of SEMDIV (denoted in gray), is highlighted in **bold**.

| Methods | Teammate 1 | | Teammate 2 | | Teammate 3 | | Teammate 4 | | Teammate 5 | | Average | |
|---|---|---|---|---|---|---|---|---|---|---|---|---|
| | R1 | R2 | R1 | R2 | R1 | R2 | R1 | R2 | R1 | R2 | R1 | R2 |
| Oracle | 0.93 | 0.96 | 0.73 | 0.76 | 0.93 | 0.86 | 0.96 | 0.92 | 0.99 | 0.98 | 0.91 | 0.90 |
| SEMDIV | $0.71_{\pm0.12}$ | $0.74_{\pm0.12}$ | $0.47_{\pm0.26}$ | $0.41_{\pm0.30}$ | $0.85_{\pm0.11}$ | $0.49_{\pm0.29}$ | $0.81_{\pm0.06}$ | $0.54_{\pm0.04}$ | $0.77_{\pm0.04}$ | $0.51_{\pm0.17}$ | $\mathbf{0.72}$ | $\mathbf{0.54}$ |
| SEMDIV-t5 | $0.63_{\pm0.16}$ | $0.68_{\pm0.16}$ | $0.01_{\pm0.01}$ | $0.00_{\pm0.00}$ | $0.61_{\pm0.06}$ | $0.17_{\pm0.08}$ | $0.63_{\pm0.11}$ | $0.23_{\pm0.25}$ | $0.66_{\pm0.10}$ | $0.33_{\pm0.20}$ | 0.51 | 0.28 |
| SEMDIV-R1 | $0.73_{\pm0.15}$ | $0.70_{\pm0.07}$ | $0.50_{\pm0.23}$ | $0.41_{\pm0.30}$ | $0.91_{\pm0.02}$ | $0.52_{\pm0.26}$ | $0.85_{\pm0.04}$ | $0.36_{\pm0.18}$ | $0.82_{\pm0.03}$ | $0.65_{\pm0.03}$ | 0.76 | 0.53 |
| SEMDIV-R2 | $0.71_{\pm0.12}$ | $0.74_{\pm0.12}$ | $0.50_{\pm0.23}$ | $0.41_{\pm0.30}$ | $0.86_{\pm0.09}$ | $0.56_{\pm0.23}$ | $0.81_{\pm0.06}$ | $0.54_{\pm0.04}$ | $0.80_{\pm0.04}$ | $0.67_{\pm0.05}$ | 0.74 | 0.58 |
| Macop-R1 | $0.28_{\pm0.04}$ | $0.00_{\pm0.00}$ | $0.32_{\pm0.02}$ | $0.00_{\pm0.00}$ | $0.75_{\pm0.06}$ | $0.15_{\pm0.03}$ | $0.82_{\pm0.03}$ | $\mathbf{0.62}_{\pm0.08}$ | $0.76_{\pm0.07}$ | $0.39_{\pm0.09}$ | 0.58 | 0.23 |
| Macop-R2 | $0.11_{\pm0.01}$ | $0.03_{\pm0.01}$ | $0.32_{\pm0.02}$ | $0.00_{\pm0.00}$ | $0.72_{\pm0.09}$ | $0.21_{\pm0.03}$ | $0.82_{\pm0.03}$ | $\mathbf{0.62}_{\pm0.08}$ | $0.74_{\pm0.09}$ | $0.40_{\pm0.08}$ | 0.54 | 0.25 |
| SEMDIV-PBT | $0.43_{\pm0.02}$ | $0.00_{\pm0.00}$ | $0.47_{\pm0.02}$ | $0.01_{\pm0.02}$ | $0.94_{\pm0.02}$ | $0.88_{\pm0.03}$ | $0.89_{\pm0.01}$ | $0.37_{\pm0.02}$ | $0.77_{\pm0.04}$ | $0.31_{\pm0.04}$ | 0.70 | 0.31 |
| Macop-PBT | $0.45_{\pm0.05}$ | $0.00_{\pm0.00}$ | $0.46_{\pm0.04}$ | $0.00_{\pm0.00}$ | $0.89_{\pm0.02}$ | $0.53_{\pm0.09}$ | $\mathbf{0.92}_{\pm0.05}$ | $0.59_{\pm0.13}$ | $\mathbf{0.88}_{\pm0.04}$ | $\mathbf{0.52}_{\pm0.13}$ | $\mathbf{0.72}$ | 0.33 |
| FCP | $0.32_{\pm0.22}$ | $0.00_{\pm0.00}$ | $0.33_{\pm0.24}$ | $0.00_{\pm0.00}$ | $0.74_{\pm0.21}$ | $0.30_{\pm0.24}$ | $0.79_{\pm0.21}$ | $0.52_{\pm0.39}$ | $0.69_{\pm0.25}$ | $0.21_{\pm0.15}$ | 0.57 | 0.21 |
| MEP | $0.45_{\pm0.03}$ | $0.00_{\pm0.00}$ | $0.47_{\pm0.02}$ | $0.00_{\pm0.00}$ | $0.95_{\pm0.01}$ | $0.91_{\pm0.02}$ | $0.88_{\pm0.04}$ | $0.31_{\pm0.09}$ | $0.77_{\pm0.02}$ | $0.32_{\pm0.05}$ | 0.70 | 0.31 |
| LIPO | $0.43_{\pm0.03}$ | $0.00_{\pm0.00}$ | $0.46_{\pm0.01}$ | $0.00_{\pm0.00}$ | $\mathbf{0.96}_{\pm0.02}$ | $\mathbf{0.92}_{\pm0.04}$ | $0.87_{\pm0.03}$ | $0.25_{\pm0.06}$ | $0.76_{\pm0.01}$ | $0.41_{\pm0.02}$ | 0.69 | 0.31 |
| LLM-Agent | $\mathbf{0.83}_{\pm0.15}$ | $\mathbf{0.85}_{\pm0.18}$ | $\mathbf{0.82}_{\pm0.09}$ | $\mathbf{0.87}_{\pm0.12}$ | $0.58_{\pm0.06}$ | $0.23_{\pm0.05}$ | $0.73_{\pm0.04}$ | $0.45_{\pm0.07}$ | $0.57_{\pm0.13}$ | $0.25_{\pm0.04}$ | 0.71 | 0.53 |

*Table 5.* Coordination performance (mean ± std) with five unseen teammates in the SMACv2 environment. "R1" and "R2" represent the success rates of task completion and agents satisfying the teammates preferred coordination behaviors, respectively. The best result in each column, excluding performance upper bounds of SEMDIV (denoted in gray), is highlighted in **bold**.

| Methods | Teammate 1 | | Teammate 2 | | Teammate 3 | | Teammate 4 | | Teammate 5 | | Average | |
|---|---|---|---|---|---|---|---|---|---|---|---|---|
| | R1 | R2 | R1 | R2 | R1 | R2 | R1 | R2 | R1 | R2 | R1 | R2 |
| Oracle | 1.00 | 1.00 | 0.96 | 0.96 | 0.98 | 0.92 | 0.82 | 0.82 | 0.96 | 0.96 | 0.94 | 0.93 |
| SEMDIV | $\mathbf{0.88}_{\pm0.10}$ | $\mathbf{0.88}_{\pm0.10}$ | $0.47_{\pm0.07}$ | $0.47_{\pm0.07}$ | $0.66_{\pm0.24}$ | $0.66_{\pm0.24}$ | $0.59_{\pm0.29}$ | $0.59_{\pm0.29}$ | $\mathbf{0.65}_{\pm0.11}$ | $\mathbf{0.59}_{\pm0.13}$ | $\mathbf{0.65}$ | $\mathbf{0.64}$ |
| SEMDIV-t5 | $0.45_{\pm0.33}$ | $0.44_{\pm0.34}$ | $0.27_{\pm0.27}$ | $0.27_{\pm0.27}$ | $0.02_{\pm0.02}$ | $0.02_{\pm0.02}$ | $0.44_{\pm0.44}$ | $0.44_{\pm0.44}$ | $0.00_{\pm0.00}$ | $0.00_{\pm0.00}$ | 0.24 | 0.23 |
| SEMDIV-R1 | $0.88_{\pm0.10}$ | $0.88_{\pm0.10}$ | $0.63_{\pm0.33}$ | $0.63_{\pm0.33}$ | $0.69_{\pm0.07}$ | $0.61_{\pm0.11}$ | $0.65_{\pm0.11}$ | $0.65_{\pm0.11}$ | $0.66_{\pm0.24}$ | $0.66_{\pm0.24}$ | 0.70 | 0.69 |
| SEMDIV-R2 | $0.88_{\pm0.10}$ | $0.88_{\pm0.10}$ | $0.63_{\pm0.33}$ | $0.63_{\pm0.33}$ | $0.69_{\pm0.07}$ | $0.61_{\pm0.11}$ | $0.65_{\pm0.11}$ | $0.65_{\pm0.11}$ | $0.66_{\pm0.24}$ | $0.66_{\pm0.24}$ | 0.70 | 0.69 |
| Macop-R1 | $0.81_{\pm0.14}$ | $0.81_{\pm0.14}$ | $0.61_{\pm0.11}$ | $0.58_{\pm0.16}$ | $0.70_{\pm0.14}$ | $0.63_{\pm0.11}$ | $0.22_{\pm0.06}$ | $0.22_{\pm0.06}$ | $0.04_{\pm0.02}$ | $0.01_{\pm0.02}$ | 0.48 | 0.45 |
| Macop-R2 | $0.81_{\pm0.14}$ | $0.81_{\pm0.14}$ | $0.61_{\pm0.11}$ | $0.58_{\pm0.16}$ | $0.70_{\pm0.14}$ | $0.63_{\pm0.11}$ | $0.22_{\pm0.06}$ | $0.22_{\pm0.06}$ | $0.03_{\pm0.02}$ | $0.02_{\pm0.02}$ | 0.47 | 0.45 |
| SEMDIV-PBT | $0.74_{\pm0.22}$ | $0.74_{\pm0.22}$ | $0.35_{\pm0.23}$ | $0.35_{\pm0.23}$ | $0.75_{\pm0.01}$ | $0.75_{\pm0.01}$ | $\mathbf{0.80}_{\pm0.20}$ | $\mathbf{0.80}_{\pm0.20}$ | $0.42_{\pm0.22}$ | $0.42_{\pm0.22}$ | 0.61 | 0.61 |
| Macop-PBT | $0.71_{\pm0.19}$ | $0.71_{\pm0.19}$ | $0.67_{\pm0.22}$ | $0.67_{\pm0.22}$ | $0.60_{\pm0.06}$ | $0.55_{\pm0.06}$ | $0.65_{\pm0.28}$ | $0.65_{\pm0.28}$ | $0.17_{\pm0.13}$ | $0.13_{\pm0.15}$ | 0.56 | 0.54 |
| FCP | $0.81_{\pm0.19}$ | $0.81_{\pm0.19}$ | $0.15_{\pm0.05}$ | $0.15_{\pm0.05}$ | $0.63_{\pm0.17}$ | $0.63_{\pm0.17}$ | $0.21_{\pm0.15}$ | $0.21_{\pm0.15}$ | $0.19_{\pm0.11}$ | $0.15_{\pm0.14}$ | 0.40 | 0.37 |
| MEP | $0.69_{\pm0.22}$ | $0.69_{\pm0.22}$ | $0.20_{\pm0.12}$ | $0.20_{\pm0.12}$ | $\mathbf{0.81}_{\pm0.08}$ | $\mathbf{0.80}_{\pm0.08}$ | $0.51_{\pm0.25}$ | $0.51_{\pm0.25}$ | $0.51_{\pm0.14}$ | $0.16_{\pm0.23}$ | 0.55 | 0.47 |
| LIPO | $0.75_{\pm0.07}$ | $0.75_{\pm0.07}$ | $0.20_{\pm0.07}$ | $0.20_{\pm0.07}$ | $0.53_{\pm0.09}$ | $0.43_{\pm0.15}$ | $0.37_{\pm0.25}$ | $0.37_{\pm0.25}$ | $0.39_{\pm0.17}$ | $0.15_{\pm0.20}$ | 0.45 | 0.38 |
| LLM-Agent | $0.30_{\pm0.04}$ | $0.30_{\pm0.04}$ | $\mathbf{0.83}_{\pm0.14}$ | $\mathbf{0.83}_{\pm0.14}$ | $0.37_{\pm0.17}$ | $0.37_{\pm0.17}$ | $0.13_{\pm0.05}$ | $0.13_{\pm0.05}$ | $0.10_{\pm0.14}$ | $0.10_{\pm0.14}$ | 0.35 | 0.35 |

*Table 6.* Coordination performance (mean ± std) with five unseen teammates in the GRF environment, where the unseen teammate coontrols player Turing. "R1" and "R2" represent the success rates of task completion and agents satisfying the teammates preferred coordination behaviors, respectively. The best result in each column, excluding performance upper bounds of SEMDIV (denoted in gray), is highlighted in **bold**.

| Methods | Teammate 1 | | Teammate 2 | | Teammate 3 | | Teammate 4 | | Teammate 5 | | Average | |
|---|---|---|---|---|---|---|---|---|---|---|---|---|
| | R1 | R2 | R1 | R2 | R1 | R2 | R1 | R2 | R1 | R2 | R1 | R2 |
| Oracle | 0.93 | 0.93 | 0.92 | 0.92 | 0.92 | 0.92 | 0.98 | 0.98 | 1.00 | 1.00 | 0.95 | 0.95 |
| SEMDIV | **0.85** $_{\pm0.17}$ | **0.85** $_{\pm0.17}$ | 0.77 $_{\pm0.15}$ | 0.77 $_{\pm0.15}$ | **0.62** $_{\pm0.44}$ | **0.62** $_{\pm0.44}$ | 0.48 $_{\pm0.08}$ | 0.48 $_{\pm0.08}$ | **0.38** $_{\pm0.38}$ | **0.35** $_{\pm0.41}$ | **0.62** | **0.61** |
| SEMDIV-t5 | 0.53 $_{\pm0.37}$ | 0.53 $_{\pm0.37}$ | 0.36 $_{\pm0.38}$ | 0.06 $_{\pm0.08}$ | 0.30 $_{\pm0.41}$ | 0.29 $_{\pm0.41}$ | 0.51 $_{\pm0.05}$ | 0.51 $_{\pm0.05}$ | 0.08 $_{\pm0.05}$ | 0.04 $_{\pm0.06}$ | 0.36 | 0.29 |
| SEMDIV-R1 | 0.85 $_{\pm0.17}$ | 0.85 $_{\pm0.17}$ | 0.92 $_{\pm0.04}$ | 0.59 $_{\pm0.42}$ | 0.81 $_{\pm0.17}$ | 0.62 $_{\pm0.44}$ | 0.93 $_{\pm0.06}$ | 0.93 $_{\pm0.06}$ | 0.90 $_{\pm0.03}$ | 0.31 $_{\pm0.43}$ | 0.88 | 0.66 |
| SEMDIV-R2 | 0.85 $_{\pm0.17}$ | 0.85 $_{\pm0.17}$ | 0.88 $_{\pm0.00}$ | 0.88 $_{\pm0.00}$ | 0.81 $_{\pm0.17}$ | 0.62 $_{\pm0.44}$ | 0.93 $_{\pm0.06}$ | 0.93 $_{\pm0.06}$ | 0.66 $_{\pm0.37}$ | 0.35 $_{\pm0.41}$ | 0.83 | 0.73 |
| Macop-R1 | 0.01 $_{\pm0.01}$ | 0.01 $_{\pm0.01}$ | **0.90** $_{\pm0.06}$ | **0.90** $_{\pm0.06}$ | 0.10 $_{\pm0.07}$ | 0.10 $_{\pm0.07}$ | **0.96** $_{\pm0.00}$ | **0.96** $_{\pm0.00}$ | 0.21 $_{\pm0.04}$ | 0.15 $_{\pm0.11}$ | 0.44 | 0.42 |
| Macop-R2 | 0.01 $_{\pm0.01}$ | 0.01 $_{\pm0.01}$ | **0.90** $_{\pm0.06}$ | **0.90** $_{\pm0.06}$ | 0.00 $_{\pm0.00}$ | 0.00 $_{\pm0.00}$ | **0.96** $_{\pm0.00}$ | **0.96** $_{\pm0.00}$ | 0.19 $_{\pm0.06}$ | 0.19 $_{\pm0.06}$ | 0.41 | 0.41 |
| SEMDIV-PBT | 0.02 $_{\pm0.03}$ | 0.02 $_{\pm0.03}$ | 0.49 $_{\pm0.28}$ | 0.49 $_{\pm0.28}$ | 0.01 $_{\pm0.01}$ | 0.00 $_{\pm0.00}$ | 0.81 $_{\pm0.15}$ | 0.81 $_{\pm0.15}$ | 0.05 $_{\pm0.02}$ | 0.05 $_{\pm0.02}$ | 0.28 | 0.27 |
| Macop-PBT | 0.00 $_{\pm0.00}$ | 0.00 $_{\pm0.00}$ | 0.32 $_{\pm0.09}$ | 0.32 $_{\pm0.09}$ | 0.00 $_{\pm0.00}$ | 0.00 $_{\pm0.00}$ | 0.93 $_{\pm0.05}$ | 0.93 $_{\pm0.05}$ | 0.05 $_{\pm0.01}$ | 0.01 $_{\pm0.02}$ | 0.26 | 0.25 |
| FCP | 0.07 $_{\pm0.07}$ | 0.07 $_{\pm0.07}$ | 0.24 $_{\pm0.01}$ | 0.24 $_{\pm0.01}$ | 0.01 $_{\pm0.01}$ | 0.00 $_{\pm0.00}$ | 0.80 $_{\pm0.04}$ | 0.80 $_{\pm0.04}$ | 0.14 $_{\pm0.10}$ | 0.12 $_{\pm0.12}$ | 0.25 | 0.25 |
| MEP | 0.20 $_{\pm0.24}$ | 0.20 $_{\pm0.24}$ | 0.24 $_{\pm0.19}$ | 0.24 $_{\pm0.19}$ | 0.05 $_{\pm0.07}$ | 0.00 $_{\pm0.00}$ | 0.61 $_{\pm0.27}$ | 0.61 $_{\pm0.27}$ | 0.12 $_{\pm0.09}$ | 0.02 $_{\pm0.00}$ | 0.24 | 0.21 |
| LIPO | 0.02 $_{\pm0.00}$ | 0.01 $_{\pm0.01}$ | 0.29 $_{\pm0.07}$ | 0.29 $_{\pm0.07}$ | 0.00 $_{\pm0.00}$ | 0.00 $_{\pm0.00}$ | 0.91 $_{\pm0.02}$ | 0.91 $_{\pm0.02}$ | 0.12 $_{\pm0.09}$ | 0.07 $_{\pm0.09}$ | 0.27 | 0.26 |
| LLM-Agent | 0.00 $_{\pm0.00}$ | 0.00 $_{\pm0.00}$ | 0.13 $_{\pm0.05}$ | 0.13 $_{\pm0.05}$ | 0.03 $_{\pm0.05}$ | 0.00 $_{\pm0.00}$ | 0.13 $_{\pm0.05}$ | 0.00 $_{\pm0.00}$ | 0.00 $_{\pm0.00}$ | 0.00 $_{\pm0.00}$ | 0.06 | 0.03 |

*Table 7.* Coordination performance (mean ± std) with five unseen teammates in the GRF environment, where the unseen teammate coontrols player Johnson. "R1" and "R2" represent the success rates of task completion and agents satisfying the teammates preferred coordination behaviors, respectively. The best result in each column, excluding performance upper bounds of SEMDIV (denoted in gray), is highlighted in **bold**.

| Methods | Teammate 1 | | Teammate 2 | | Teammate 3 | | Teammate 4 | | Teammate 5 | | Average | |
|---|---|---|---|---|---|---|---|---|---|---|---|---|
| | R1 | R2 | R1 | R2 | R1 | R2 | R1 | R2 | R1 | R2 | R1 | R2 |
| Oracle | 0.93 | 0.93 | 0.92 | 0.92 | 0.92 | 0.92 | 0.98 | 0.98 | 1.00 | 1.00 | 0.95 | 0.95 |
| SEMDIV | 0.78 $_{\pm0.09}$ | **0.29** $_{\pm0.41}$ | 0.79 $_{\pm0.12}$ | 0.79 $_{\pm0.12}$ | **0.90** $_{\pm0.00}$ | **0.90** $_{\pm0.00}$ | 0.65 $_{\pm0.10}$ | 0.65 $_{\pm0.10}$ | 0.52 $_{\pm0.13}$ | **0.51** $_{\pm0.15}$ | 0.73 | **0.63** |
| SEMDIV-t5 | 0.32 $_{\pm0.19}$ | 0.16 $_{\pm0.23}$ | 0.61 $_{\pm0.38}$ | 0.61 $_{\pm0.38}$ | 0.87 $_{\pm0.03}$ | 0.87 $_{\pm0.03}$ | 0.53 $_{\pm0.06}$ | 0.53 $_{\pm0.06}$ | 0.59 $_{\pm0.25}$ | 0.10 $_{\pm0.14}$ | 0.58 | 0.45 |
| SEMDIV-R1 | 0.85 $_{\pm0.05}$ | 0.30 $_{\pm0.41}$ | 0.90 $_{\pm0.03}$ | 0.90 $_{\pm0.03}$ | 0.93 $_{\pm0.01}$ | 0.93 $_{\pm0.01}$ | 0.81 $_{\pm0.03}$ | 0.81 $_{\pm0.04}$ | 0.94 $_{\pm0.03}$ | 0.00 $_{\pm0.00}$ | 0.89 | 0.59 |
| SEMDIV-R2 | 0.49 $_{\pm0.32}$ | 0.49 $_{\pm0.32}$ | 0.90 $_{\pm0.03}$ | 0.90 $_{\pm0.03}$ | 0.93 $_{\pm0.01}$ | 0.93 $_{\pm0.01}$ | 0.81 $_{\pm0.03}$ | 0.81 $_{\pm0.04}$ | 0.58 $_{\pm0.04}$ | 0.58 $_{\pm0.04}$ | 0.74 | 0.74 |
| Macop-R1 | 0.62 $_{\pm0.03}$ | 0.01 $_{\pm0.01}$ | 0.72 $_{\pm0.01}$ | 0.72 $_{\pm0.01}$ | 0.85 $_{\pm0.02}$ | 0.85 $_{\pm0.02}$ | 0.77 $_{\pm0.02}$ | 0.77 $_{\pm0.02}$ | 0.76 $_{\pm0.01}$ | 0.01 $_{\pm0.01}$ | 0.74 | 0.47 |
| Macop-R2 | 0.46 $_{\pm0.10}$ | 0.05 $_{\pm0.03}$ | 0.72 $_{\pm0.01}$ | 0.72 $_{\pm0.01}$ | 0.85 $_{\pm0.02}$ | 0.85 $_{\pm0.02}$ | 0.77 $_{\pm0.02}$ | 0.77 $_{\pm0.02}$ | 0.76 $_{\pm0.01}$ | 0.04 $_{\pm0.04}$ | 0.71 | 0.49 |
| SEMDIV-PBT | **0.92** $_{\pm0.04}$ | 0.00 $_{\pm0.00}$ | **0.87** $_{\pm0.06}$ | **0.87** $_{\pm0.06}$ | 0.85 $_{\pm0.04}$ | 0.85 $_{\pm0.04}$ | 0.83 $_{\pm0.03}$ | 0.83 $_{\pm0.03}$ | **0.87** $_{\pm0.05}$ | 0.01 $_{\pm0.01}$ | **0.87** | 0.51 |
| Macop-PBT | 0.61 $_{\pm0.04}$ | 0.01 $_{\pm0.01}$ | 0.68 $_{\pm0.06}$ | 0.66 $_{\pm0.09}$ | 0.81 $_{\pm0.05}$ | 0.80 $_{\pm0.07}$ | 0.74 $_{\pm0.01}$ | 0.71 $_{\pm0.05}$ | 0.74 $_{\pm0.02}$ | 0.01 $_{\pm0.01}$ | 0.72 | 0.44 |
| FCP | 0.72 $_{\pm0.13}$ | 0.00 $_{\pm0.00}$ | 0.55 $_{\pm0.26}$ | 0.55 $_{\pm0.26}$ | 0.89 $_{\pm0.01}$ | 0.89 $_{\pm0.01}$ | **0.91** $_{\pm0.07}$ | **0.91** $_{\pm0.07}$ | 0.64 $_{\pm0.17}$ | 0.00 $_{\pm0.00}$ | 0.74 | 0.47 |
| MEP | 0.63 $_{\pm0.28}$ | 0.02 $_{\pm0.02}$ | 0.79 $_{\pm0.03}$ | 0.79 $_{\pm0.03}$ | **0.90** $_{\pm0.01}$ | **0.90** $_{\pm0.01}$ | 0.70 $_{\pm0.15}$ | 0.70 $_{\pm0.15}$ | 0.78 $_{\pm0.05}$ | 0.00 $_{\pm0.00}$ | 0.76 | 0.48 |
| LIPO | 0.60 $_{\pm0.32}$ | 0.00 $_{\pm0.00}$ | 0.72 $_{\pm0.12}$ | 0.72 $_{\pm0.12}$ | 0.86 $_{\pm0.02}$ | 0.86 $_{\pm0.02}$ | 0.80 $_{\pm0.06}$ | 0.80 $_{\pm0.06}$ | 0.75 $_{\pm0.07}$ | 0.01 $_{\pm0.01}$ | 0.75 | 0.48 |
| LLM-Agent | 0.07 $_{\pm0.09}$ | 0.07 $_{\pm0.09}$ | 0.10 $_{\pm0.08}$ | 0.10 $_{\pm0.08}$ | 0.80 $_{\pm0.00}$ | 0.80 $_{\pm0.00}$ | 0.10 $_{\pm0.08}$ | 0.00 $_{\pm0.00}$ | 0.07 $_{\pm0.09}$ | 0.07 $_{\pm0.09}$ | 0.23 | 0.21 |

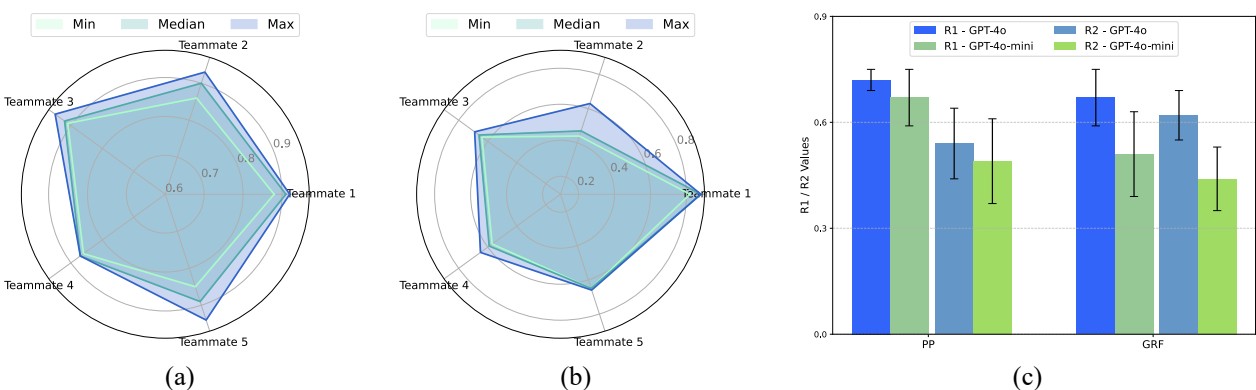

*Figure 7.* Experiments on the impact of ambiguity in teammates' coordination behaviors and the quality of LLMs. (a)(b) The minimum, median, and maximum R1 values of each teammate's 10 different behavior descriptions in LBF and SMACv2. (c) The performance of SEMDIV when using different LLMs in PP and GRF.

### E.2. The Impact of Ambiguity in Teammates' Coordination Behaviors

In previous experiments, each testing teammate was described using a single, unambiguous statement for clarity. In this section, we investigate the impact of introducing ambiguity into these descriptions and evaluate the robustness of SEMDIV. Specifically, for each testing teammate, we input its original description and the task information into an LLM, prompting it to generate 9 alternative phrasings of the original description. This process yields a total of 10 descriptions for each teammate. We then evaluate SEMDIV using all 10 descriptions for each teammate, calculating the minimum, median, and maximum R1 values across these variations in the LBF and SMACv2 environments. As shown in Figure 7(a)(b), the performance of SEMDIV remains consistent despite the introduced ambiguity, demonstrating the robustness of its language-based reasoning process for head selection.

### E.3. The Impact of the Quality of LLMs

LLMs play a critical role in the design of SEMDIV. To assess their impact, we replace the `gpt-4o-2024-08-06` model with `gpt-4o-mini` and conduct experiments in the PP and GRF environments. As illustrated in Figure 7(c), the use of GPT-4o-mini results in a modest performance decline, demonstrating that even a smaller LLM can effectively support SEMDIV 's functionality. With the ongoing development of more advanced LLMs offering enhanced capabilities (OpenAI, 2024; DeepSeek-AI, 2025), SEMDIV holds the potential for further performance improvements.

## F. Prompt Engineering

In this section, we provide the prompts for LLMs used in this paper.

### F.1. Task Information

We first provide the prompts about task information across all four environments, as they are frequently reused in prompts for different purposes in SEMDIV.

**LBF**:

> You are an expert in cooperative multi-agent reinforcement learning (MARL) and code generation. We are going to train a team of two players in the Level-Based Foraging (LBF) game. The game is a 2D square grid-world with two agents, and four foods (denoted as food "A", "B", "C", and "D") are scattered in four different corners. Each player controls an agent. They need to choose a same food and move towards it, and be at adjacent grids of it together to collect the food. When agents successfully collect the first food, like food "B", they get reward 1 and the game ends.
>
> Here's a part of the original code:
>
> ```python
> class ForagingEnv(Env):
>     self.agents_position : {"1": np.ndarray[(2,)], "2": np.ndarray[(2,)]}
>     self.foods_position : {"A": np.array([0, 0]), "B": np.array([0, 7]), "C": np.array([7, 0]), "D": np.array([7, 7])}
>     self.collected_food : str # record the food ("A" / "B" / "C" / "D" / "") collected by the team, and "" means no food has
>     ↪ been collected yet.
>     # other attributes and functions
>
>     def agent_food_distance(self, agent_idx: str, food_idx: str):
>         agent_pos = self.agents_position[agent_idx]
>         food_pos = self.foods_position[food_idx]
>         distance = np.linalg.norm(agent_pos - food_pos)
>         return distance
>
>     def step(self):
>         # other codes
>         reward = 0
>         # process collectings: if agents successfully collect one food, reward = 1
>         for food, (food_row, food_col) in self.foods_position.items():
>             # 2 agents be at adjacent grids of it together to collect the food
>             n_adj_players = self.adjacent_player_number(food_row, food_col)
>             if n_adj_players == 2:
>                 self.collected_food = food
>                 reward = 1
>                 break
>         # when agents successfully collect a food, they get reward = 1 and the game ends.
>         done = (reward == 1) or (self.current_step >= self._max_episode_steps)
>         reward += self.additional_reward()
>         # return new state, reward, done, and other step info
> ```

**PP**:

> You are an expert in cooperative multi-agent reinforcement learning (MARL) and code generation. We are going to train a team of two players in the Predator-Prey (PP) game. The game is a 2D world with two predators and five prey (two stags S1 S2 and three rabbits R1 R2 R3). Each player controls a predator. They need to choose the prey to catch (like S1 or R2+R3), then chase the chosen prey to catch them. Stags require two predators to catch at the same time. If only one predator is near them, both players will be punished. Rabbits only require one predator to catch them. When players successfully catch a stag, they get reward 1. When players successfully catch a rabbit, they get reward 0.5.
>
> Here's a part of the original code:
>
> ```python
> class Game:
>     self.predators_position : {"1": np.ndarray[(2,)], "2": np.ndarray[(2,)]} # Initialization: both np.random.uniform(-0.1,
>     ↪ +0.1, 2)
>     self.prey_position : {"S1": np.ndarray[(2,)], "S2": np.ndarray[(2,)], "R1": np.ndarray[(2,)], "R2": np.ndarray[(2,)], "R3":
>     ↪ np.ndarray[(2,)]} # Initialization: "S1": [1., 0.], "S2": [-1., 0.], "R1": [0.8, 0.6], "R2": [-0.8, 0.6], "R3": [0.,
>     ↪ -1.]
>     self.caught_prey_set = set() # record the prey caught by the team, like {"S1"} or {"R2", "R3"}, and an empty set means no
>     ↪ prey has been caught yet.
>     def entity_distance(self, entity1 : str, entity2 : str) -> float:
>         # return the distance between the input entities, like "1" and "2", "1" and "S1", "R1" and "R2", etc.
>     def get_prey_level(self, prey : str) -> int:
>         # return 2 for "S1" and "S2", return 1 for "R1" and "R2" and "R3"
>     def get_num_predator_nearby(self, prey : str) -> int:
>         # return the number of predators near / catching the prey (distance <= 0.25), can be 0 or 1 or 2
>     def step(self):
>         # other codes that change positions
>         reward = 0.0
>         for prey in self.prey_position.keys():
>             prey_level = self.get_prey_level(prey)
>             num_predator_nearby = self.get_num_predator_nearby(prey)
>             if num_predator_nearby == 0:
>                 continue
>             elif 0 < num_predator_nearby < prey_level:
>                 reward -= 0.01
>             if num_predator_nearby >= prey_level:
>                 reward += prey_level / 2
>                 self.caught_prey_set.add(prey)
>         reward += self.additional_reward()
>         # other codes
> ```

**SMACv2**:

> You are an expert in cooperative multi-agent reinforcement learning (MARL) and code generation. We are going to train a team of two players in the Starcraft Multi-Agent Challenge (SMAC) game, which involves unit micromanagement tasks. In this game, ally units need to beat enemy units controlled by the built-in AI. Specifically, each player controlls a marine agent ("1" and "2") to beat four enemy marines ("A", "B", "C", and "D"). The two marine agents are spawned at the center of the field, and four enemies are scattered in four different corners. Agents need to choose a same enemy, move towards it, and fire at it together to kill it. When agents successfully kill the first enemy, like enemy "B", they get a reward about 10 and the game ends. If both agents are killed, they lose.
>
> Here's a part of the original code:
>
> ```python
> class Game:
>     self.agents_position : {"1": np.ndarray[(2,)], "2": np.ndarray[(2,)]}
>     self.enemies_position : {"A": np.ndarray[(2,)], "B": np.ndarray[(2,)], "C": np.ndarray[(2,)], "D": np.ndarray[(2,)]}
>     # these 2D positions are calculated as [(x - self.center_x) / self.max_distance_x, (y - self.center_y) /
>     ↪ self.max_distance_y]
>     # initial positions: agents near [0., 0.], "A" lower left, "B" upper left, "C" upper right, "D" bottom right
>     # for agents and enemies that are killed, their postions will be set to [0., 0.]
>     self.killed_enemy : str # record the enemy ("A" / "B" / "C" / "D" / "") killed by the team, and "" means no enemy has been
>     ↪ killed yet.
>     # other attributes and functions
>
>     def agent_enemy_distance(self, agent_idx: str, enemy_idx: str):
>         agent_pos = self.agents_position[agent_idx]
>         enemy_pos = self.enemies_position[enemy_idx]
>         distance = np.linalg.norm(agent_pos - enemy_pos)
>         return distance
>
>     def step(self):
>         reward = 0.0
>         # other codes that change the battle state the above attributes, and calculate the original reward
>         reward += self.additional_reward()
>         # other codes
> ```

**GRF**:

You are an expert in cooperative multi-agent reinforcement learning (MARL) and code generation. We are going to train a team of two football players (Turing and Johnson) in the Google Research Football (GRF) game. They try to score from the edge of the box, Johnson is on the side with the ball, Turing is at the center and facing the goalkeeper (Meitner). Our team gets reward 1 when scoring a goal. An episode ends when our team scores a goal, or Meitner owns the ball, or the ball is out of bounds.

Here's a part of the original code:

```python
class Game:
    ## 1. Location information
    # The closer to the opponent's goal, the larger the x-coordinate. The y-coordinate of the left half of the field is < 0,
    ↪   and the y-coordinate of the right half is > zero.
    self.ball_position : np.ndarray[(2,)] # ball's (x, y) coordinate, (0.7, -0.28) at the beginning
    self.Turing_position : np.ndarray[(2,)] # Turing's (x, y) coordinate, (0.7, 0.0) at the beginning
    self.Johnson_position : np.ndarray[(2,)] # Johnson's (x, y) coordinate, (0.7, -0.3) at the beginning
    self.Meitner_position : np.ndarray[(2,)] # Meitner' (x, y) coordinate, (1.0, 0.0) at the beginning
    # Coordinates of the lower left and right corners of the goal are about (1.0, -0.04) and (1.0, 0.04)
    ## 2. Critical game-level information
    self.pass_history : list # List to store the history of passes as tuples, with the first element as the player who made the
    ↪   pass and the second element as the player who received it, for example, [("Johnson", "Turing"), ("Turing", "Johnson")]
    self.score : bool # True if the team scores a goal at this step and False otherwise
    self.score_Turing : bool # True if Turing scores a goal at this step and False otherwise
    self.score_Johnson : bool # True if Johnson scores a goal at this step and False otherwise
    def step(self):
        # other codes that change the above attributes
        reward = 0.0
        if self.score:
            reward += 1
        reward += self.additional_reward()
        # other codes
```

## F.2. Behavior Generator

We provide the prompt for the LLM behavior generator.

{Task information}

Human player teams may have specific cooperation preferences to play the game. They have their own additional_reward shown in the code. A new player outside a team needs to learn and adapt these preferences to cooperate well after joining the team.

Here are some behavior examples:

- Example 1: {A previous valid coordination behavior}

Based on the information above, think step by step to come up with another possible cooperation preference. The preference should be deterministic and concrete. It should be as simple as possible. Avoid conditional terms like if, unless, when, etc. Avoid sequential behaviors like "first X, then Y". It should be easily implemented in python codes using the provided code snippet. It should not conflict with the original task objective.

Finally, output the preference in the format: "Human players may prefer to {preference}".

## F.3. Reward Generator

We provide the prompt for the LLM reward generator.

{Task information}

Now we want to train a team with this specific cooperation behavior: {A natural language coordination behavior}

According to this cooperation preference, write an operational and executable reward function that formats as "def additional_reward(self) → float" and returns the "reward : float" only.

1. Please think step by step and tell us what this code means. 2. The code function must align with the cooperation

preference. 3. It can be a dense reward that guides the team to learn the cooperation preference. 4. Short and simple code is better.

We have tried some reward function code before, but they are not good enough:

Attempt 1: {Information of the previous attempt}

Based on these information, You may consider change or rewrite the function.

## F.4. Alignment Verification

We provide the prompt for the alignment examination between behaviors and policies.

{Task information}

We tried to train a team with this specific cooperation behavior: {A natural language coordination behavior}

After training the team with this reward function, we ran it for multiple episodes: {episode information}

Based on the information above, please review if the running behavior of the team aligns with the desired behavior or not. Think step by step, and tell us your answer. Make sure your output contains a string "::1::" if your answer is "Yes" and contains a string "::0::" if your answer is "No".

## F.5. Policy Selector

We provide the prompt for the head selection process before testing with an unseen teammate in GRF as an example.

You are an expert in football. We are going to build a team of two football players (Turing and Johnson, no other teammates). They need to score from the edge of the box. When the game starts, Johnson is on the left side controlling the ball, Turing is at the center and facing the goalkeeper.

Johnson was trained under the same situation, but with different teammates other than Turing to achieve the following cooperation preferences, and learned corresponding policies:

1: {learned behavior 1}, ... ,
6: {learned behavior 6}

Now, Turing says that: "{testing teammates' coordination behavior}" Based on the information above, please carefully analyze the game, the ball, the policies, etc. Think step by step to select the policy (1~6) for Johnson that can best coordinate with Turing and satisfy his preferences. Output your answer in the format "[n]". For example, if your answer is policy 3, output "[3]".

## F.6. The LLM-Agent Baseline

Here we provide the prompts for the LLM-Agent baseline in all four environments.

**LBF**:

You are an expert in the Level-Based Foraging (LBF) game. We are going to build a team of two players 1 and 2 in the LBF game. They need to collect a food together. There are four foods (A, B, C, D) in the field. foods_position : A = [0, 0], B = [0, 7], C = [7, 0], D = [7, 7] Before the game starts, player 2 said that: "I prefer to collect food A." Now, player 1 is at [4. 3.], player 2 is at [3. 4.]. Based on the information above, select an action (0~4) for player 1 that can best coordinate with player 2 and satisfy his preferences Output your answer in the format "[n]". For example, if your answer is action 3, output "[3]".

Actions: 0 (no-op), 1 (x coordinate minus one), 2 (x coordinate plus one), 3 (y coordinate minus one), 4 (y coordinate plus one).

Output:

**PP**:

You are an expert in the predator-prey (PP) game. We are going to build a team of two players 1 and 2 controlling two predators in the PP game. They need to chase and catch the prey. There are five prey including two stags (S1, S2) and three rabbits (R1, R2, R3). Stags require two predators to catch at the same time. If only one predator is near them, both players will be punished. Rabbits only require one predator to catch them. Before the game starts, player 2 said that: "I prefer to catch stag S1." Now, player 1 is at (0.06, 0.02), player 2 is at (-0.09, -0.02), S1 is at (1.00, 0.00), S2 is at (-1.00, 0.00), R1 is at (0.80, 0.60), R2 is at (-0.80, 0.60), R3 is at (0.00, -1.00). Based on the information above, select an action (0∼4) for player 1 that can best coordinate with player 2 and satisfy his preferences. Output your answer in the format "[n]". For example, if your answer is action 3, output "[3]".

Actions: 0 (no-op), 1 (accelerate towards +x direction), 2 (accelerate towards -x direction), 3 (accelerate towards +y direction), 4 (accelerate towards -y direction).

Output:

**SMACv2**:

You are an expert in the Starcraft Multi-Agent Challenge (SMAC) game. We are going to build a team of two players 1 and 2 controlling two marines in the SMAC game. They need to beat enemy units controlled by the built-in AI. There are four enemy marines (A, B, C, and D) scattered in four different corners. Initial positions: agents near the map center [16., 16.], enemy A at the lower left corner, B at the upper left corner, C at the upper right corner, D at the bottom right corner. Agents need to choose a same enemy, move towards it, and fire at it together to kill it. When agents successfully kill the first enemy, like enemy B, they win. If both agents are killed, they lose. Before the game starts, player 2 said that: "I prefer to kill enemy C." Now, player 1 is at [16.95, 22.19], with health value 1.00 (1 is full health, 0 is dead). Player 2 is at [17.15, 21.50], with distance 0.72 and health value 0.60. Enemy A is out of sight or dead, so cannot be observed and attacked. Enemy B is out of sight or dead, so cannot be observed and attacked. Enemy C is out of sight or dead, so cannot be observed and attacked. Enemy D is out of sight or dead, so cannot be observed and attacked. Based on the information above, please select an action (1∼9) for player 1 that can best coordinate with player 2 and satisfy his preferences. Output your answer in the format "[n]". For example, if your answer is action 3, output "[3]".

Actions: 1 (stop the current action), 2 (move north), 3 (move south), 4 (move east), 5 (move west), 6 (shoot enemy A), 7 (shoot enemy B), 8 (shoot enemy C), 9 (shoot enemy D).

Available actions: [1, 2, 3, 4, 5, 8].

Output:

**GRF**:

You are an expert in football. A team of two football players (Turing and Johnson) need to score from the edge of the box. When the game starts, Johnson is on the side controlling the ball, Turing is at the center and facing the goalkeeper Meitner. Before the game, Turing said that: "I prefer Johnson to score." Now, the ball is at [0.70 -0.28] with direction [0.00 -0.00], Turing is at [0.70 -0.01], Johnson is at [0.70 -0.29], Meitner is at [0.99 -0.02]. The center point of the goal is at [1.0, 0.0]. For [x, y] coordinates, -x direction is on the left, +x direction is on the right, -y direction is on the top, +y direction is on the bottom. The pass history of our team is []. Based on the information above, please select an action (1∼18) for Johnson that can best coordinate with Turing and satisfy his preferences. Output your answer in the format "[n]". For example, if your answer is action 3, output "[3]".

Actions: 1 (run to the left), 2 (run to the top-left), 3 (run to the top), 4 (run to the top-right), 5 (run to the right), 6 (run to the bottom-right), 7 (run to the bottom), 8 (run to the bottom-left), 9 (perform a long pass), 10 (perform a high pass), 11 (perform a short pass), 12 (perform a shot), 13 (start sprinting), 14 (reset current movement direction), 15 (stop sprinting), 16 (perform a slide), 17 (start dribbling), 18 (stop dribbling),

Output:

