# OpenReview forum: "LLM-Assisted Semantically Diverse Teammate Generation for Efficient Multi-agent Coordination"
_ICML.cc/2025/Conference — ICML 2025 poster_

### Official Review · Reviewer_Q2F9 · 2025-03-09

**Overall Recommendation:** 5

**Summary:**

This work proposes a novel algorithm called “SemDiv” which uses LLMs to generate semantically diverse behavior in MARL. Specifically, SemDiv uses an LLM to (1) generate a language description of a plausible, novel coordination behavior, (2) generate a reward function that incentivizes that coordination behavior, and (3) validate that trained agents actually follow that behavior. SemDiv also generates a common “coordinating agent” that can coordinate with unseen partners given a language description of their convention. Experiments in multiple standard MARL environments validate the performance of SemDiv agents when paired with unseen partners.

## Update After Rebuttal

The author provided satisfactory answers to my questions, so I am comfortable keeping my review as a strong accept.

**Claims And Evidence:**

The claims made in the paper are supported by clear and convincing evidence.

**Essential References Not Discussed:**

- The problem formulation seems to follow the N-AHT from “N-Agent Ad Hoc Teamwork” in NeurIPS 2024
- “Language Instructed Reinforcement Learning for Human-AI Coordination” in ICML 2023 provides an alternative way of regularizing a policy given a language instruction (InstructRL)
- “Adversarial Diversity in Hanabi” in ICLR 2023 and “Diverse Conventions for Human-AI Collaboration” in NeurIPS 2023 are additional techniques for cross-play based diversity (ADVERSITY and CoMeDi)

**Experimental Designs Or Analyses:**

For completeness, it would be good to show the performance of FCP, MEP, and LIPO with the multi-head network architecture (i.e. keep stage 1 the same but replace stage 2 with the technique described in section 3.3)

**Methods And Evaluation Criteria:**

The methods and evaluation make sense for the problem at hand (though I have specific questions about alternative approaches in the “questions for authors” section of my review)

**Other Comments Or Suggestions:**

- It was initially unclear to me whether section 3.3 was presented as a novel contribution of this work or identical to prior work. I think this needs to be clarified in the text (i.e. rephrase the first sentence of the second paragraph in 3.3), since I was initially wondering if this is different from what Macop does when reading the results section.
- The “best results” in tables should also include results where the confidence intervals overlap. For instance, LLM-Agent seems to be within the margin of error for many of the results.
- “Incorporate” -> “incorporating” on second column of line 320

**Other Strengths And Weaknesses:**

I consider this to be a truly revolutionary paper for the MARL field. The presented pipeline for generating a diverse set of teammates is broadly applicable to many multi-agent settings and enables many real-world applications for more personalized AI co-agents.

Weaknesses:
- There aren't any human user studies. It is currently a bit unclear how directly the presented results transfer when working with humans.
- This paper also lacks ablations over other potential options for generating single teammates given prompts, namely InstructRL.
- Due to the dependence on LLMs, it is unclear whether the presented technique can learn strategies in complex games that require deeper understanding of game mechanics before planning coordination strategies.

**Questions For Authors:**

- Why is a regularization term used to constrain updates in section 3.3 instead of using a more standard multi-task RL setup? I.e. first generate the diverse training population and then generate the coordinating agent by randomly sampling partners
    - Furthermore, why isn’t behavior cloning used, considering that the complementary policy already exists from the training regime? The complementary policy should already be a strong coordinator, so conducting RL again from scratch seems unnecessary; a QDagger-like approach such as the one from CoMeDi may be helpful.
- How are the testing teammates generated given the descriptions? Specifically, does it follow the same pipeline as your technique (skipping section 3.1 only), is it a scripted agent, or is it using a hand-designed reward function?
- How are R2 values calculated exactly? The definition of “satisfying teammates’ preferred coordination behaviors” is vague.

**Relation To Broader Scientific Literature:**

The key contribution of this work is the pipeline of (1) generating a language description of a coordination behavior, (2) converting this description into a policy, and (3) iterating to ensure novelty and diversity.
- Other works have addressed the middle stage of the pipeline (“InstructRL” referenced in the next section of the review). However, the pipeline as a whole is still novel.

**Theoretical Claims:**

N/A

---

> ### Author Rebuttal · Authors · 2025-04-01
>
> We sincerely appreciate your recognition and the valuable comments! Extra experimental results can be found in this [link](https://telling-floor-898.notion.site/1c7c2fed721a80b9ba7ef7fa2b3bffed).
>
> **Q1: The performance of population-based baselines with the multi-head network architecture.**
>
> A: We train the agents using a multi-head architecture (denoted as {}-mh) with the generated teammates. For testing with unseen teammates, we evaluate the baselines by selecting the best-performing head among all learned heads (the same as the -R1 variants in the paper), which requires extensive interactions. As shown in Table 11 in the link. the multi-head architecture significantly improves performance in LBF but yields no gains in the more complex SMACv2 environment. In both cases, a substantial performance gap remains between the baselines and SemDiv, highlighting the necessity of semantic-level diverse teammate generation.
>
> **Q2: Discuss essential references.**
>
> A: Thanks for pointing out these essential related work. We will add the discussion in the paper.
>
> - The N-AHT framework [1] does align with our problem formulation, and we extend it by introducing natural language descriptions about testing teammates, making it more suitable for multi-agent systems with communication and intention sharing.
> - InstructRL [2] is a notable work that leverages LLMs to guide RL by using their decisions to regularize policy learning. This approach offers an alternative to reward-generation methods like those discussed in our paper. However, InstructRL requires querying the LLM at every step, which is computationally expensive and limits its applicability to relatively simple tasks. Due to these constraints, we did not include it as a baseline in our study.
> - ADVERSITY [3] and CoMeDi [4] propose novel techniques for cross-play based diversity, which can be categorized into the “policy-level” methods in our paper and introduced as strong baselines.
>
> **Q3: Human user studies and more complex games.**
>
> A: Both directions present valuable opportunities, particularly in bridging the gap between simulated coordination and human-AI teamwork, as well as extending the framework to richer strategic environments. We plan to investigate these challenges in future work.
>
> **Q4: Design choices for continual learning.**
>
> A: We adopt a continual learning paradigm instead of a two-stage framework (like FCP or MEP) because it allows us to leverage previously generated grounded behaviors as positive examples, guiding the LLMs toward progressively better behavior generation. In contrast, a two-stage approach, where behaviors are generated first and then used to train the teammate population, isolates the learning process of different teammates and may reduce diversity.
>
> **Q5: Design choices for agents’ RL instead of BC.**
>
> A: While behavior cloning (BC) from the complementary policy is a viable approach, we choose RL for two key reasons: 1. Surpassing complementary policy performance: BC merely imitates the complementary policy, capping performance at the quality of the training data. In contrast, RL enables exploration and optimization beyond demonstrations, potentially discovering superior coordination strategies. 2. Mitigating distribution shift: BC may struggle with OOD teammates due to its inability to recover from unfamiliar situations. Since our focus is on generalizing to unseen teammates, BC’s reliance on static datasets may lead to failure.
>
> We apologize for unclear expressions and typos, and will correct them in the paper.
>
> - Relationship between Section 3.3 and Macop: For training, we remove the head merging technique of Macop, since the novelty of new teammates if confirmed in the policy verification process. For testing, we utilize language-based reasoning to select optimal heads, avoiding Macop’s need for trial-and-error interactions to improving efficiency. We will rephrase the first sentence of the second paragraph in Section 3.3: To address these challenges, SemDiv adopts a multi-head network architecture \cite{owl,macpro} similar with Macop \cite{macop} and empowers the agents with continual learning ability.
> - We report the 95% confidence intervals (CI) of SemDiv and the next best performing method (Macop-R1) in the link.
> - Testing teammate generation: We train these teammates using hand-designed reward functions and manually verify that they exhibit the desired behaviors. These reward functions are simple and sparse (returning either 1 or 0), allowing for direct derivation of the R2 values.
>
> References
>
> [1] Wang et al. N-Agent Ad Hoc Teamwork. NeurIPS 2024.
>
> [2] Hu et al. Language Instructed Reinforcement Learning for Human-AI Coordination. ICML 2023.
>
> [3] Cui et al. Adversarial Diversity in Hanabi. ICLR 2023.
>
> [4] Sarkar et al. Diverse Conventions for Human-AI Collaboration. NeurIPS 2023.

---

### Official Review · Reviewer_rgCD · 2025-03-09

**Overall Recommendation:** 3

**Summary:**

The paper introduces SEMDIV, a novel framework that uses LLMs to generate semantically diverse teammates for efficient multi-agent coordination. Unlike traditional methods that focus on policy-level diversity, SEMDIV iteratively generates natural language descriptions of coordination behaviors, translates them into reward functions, and trains teammates to embody these behaviors. This approach allows agents to learn and adapt through continual learning with a multi-head architecture, selecting the most suitable policy through language-based reasoning. Experiments across four environments show SEMDIV’s superior performance in coordinating with unseen teammates.

**Claims And Evidence:**

Yes

**Essential References Not Discussed:**

no

**Experimental Designs Or Analyses:**

This work conducts experiments in four multi-agent scenarios. However, several issues remain:
- The number of agents in these scenarios is very limited — only two — which raises concerns about the scalability of the proposed method. It would be more comprehensive to evaluate additional tasks in SMACv2 and GRF with a greater number of agents.
- All selected scenarios involve a discrete action space; it would be beneficial to assess performance in tasks with a continuous action space, such as MAMuJoCo.
- The diversity of testing teammates appears to be limited. Including a wider variety of teammates, such as lazy agents, would provide a more thorough evaluation.

**Methods And Evaluation Criteria:**

Yes, this work focuses on the multi-agent domain with diverse teammates, and the proposed method and evaluation criteria are well-suited to this context.

**Other Comments Or Suggestions:**

1. I am curious about the scaling effect of SEMDIV with a larger number of teammates.
2. I am unclear about the meaning of SEMDIV-R1/R2. Does it refer to evaluating each heading and selecting the best one?
3. minor: It would be helpful to add detailed explanations to the figures, such as Figures 1 and 2.
4. Typos: Second paragraph of Section 4.1 — "In each environment, we train **five** teammates exhibiting distinct and representative coordination behaviors." Should this be six teammates?

**Other Strengths And Weaknesses:**

Strengths
--

1. The paper is well-structured and the idea is well-motivated.
2. The implementation details and prompt designs are clearly and thoroughly described.
3. The experimental results are compelling and thoroughly explained.

Weaknesses
--

1. The proposed method heavily relies on several assumptions about the environments and testing setups, including (i) accessible attributes and APIs of the environments, (ii) language descriptions of episodes, and (iii) language descriptions of teammates' behavior during execution, which may not hold in other scenarios.
2. I find Section 3.2 a little bit confusing: (i) What does $\tilde{\pi}_m^{tm}$ refer to during teammate policy training? (ii) Why are both $\lambda$ configurations necessary in Equation (2)?
3. There are concerns regarding the experimental design, as noted above.
4. The discussion of limitations is insufficient.

**Questions For Authors:**

How does SEMDIV handle contradictory behaviors across teammates? Could conflicting policies cause catastrophic forgetting despite the regularization term?

**Relation To Broader Scientific Literature:**

The work integrates LLMs into the multi-agent RL, extending research on ad-hoc teamwork and zero-shot coordination. It thoughtfully builds on existing diversity-inducing methods while addressing their limitations on semantic information.

**Theoretical Claims:**

There is not theoretical claim in this submission.

---

> ### Author Rebuttal · Authors · 2025-04-01
>
> We sincerely thank you for the valuable comments! Extra experimental results can be found in this [link](https://telling-floor-898.notion.site/1c7c2fed721a80b9ba7ef7fa2b3bffed).
>
> **Q1: Experiments with more agents and with continuous action space.**
>
> A: SemDiv is agnostic to team sizes and action spaces, still achieving the best performance compared with baselines, indicating its scalability and potential to solve more complex tasks. The results are in Table 8 in the link.
>
> First, we extend the original LBF task to a 3-agent setting, where it takes 3 agents to collect a food at the same step. During testing, one unseen agent joins the team. Training details (like MARL methods, prompts for LLMs, etc), and behaviors of testing teammates, are similar with the 2-agent experiments.
>
> Next, we include the Spread task based on the HARL codebase [1], in which 3 agents with continuous action space need to approach three different landmarks. For training, we select MAPPO similar with GRF. For LLM prompts, we slightly modify the ones used in PP since both of them are MPE tasks. During testing, 6 unseen teammates trained in teams with different approaching strategies (different agent-landmark pairs) are joined.
>
> **Q2: Testing with lazy agents.**
>
> A: We build lazy agents that take the noop action or move back and forth in the initial position, in all four environments used in the paper. We set their descriptions as “I am a lazy agent and I will do nothing” when testing SemDiv. The results are shown in Table 9.
>
> - LBF & SMACv2: All methods fail as ≥2 agents are needed for food/enemy tasks.
> - PP: Baselines match SemDiv by ignoring lazy teammates (reward for catching a rabbit = 0.5).
> - GRF: SemDiv excels by learning attack strategies without passing and selecting these policy heads.
>
> **Q3: Discuss several assumptions and limitations.**
>
> A: (1) Environment APIs, and language descriptions of episodes. Both assumptions are important considerations, which are well-established in prior LLM-assisted RL research [2,3,4]. However, we acknowledge that relaxing these assumptions is valuable future work. (2) Language descriptions of teammates' behavior during execution. Please refer to Q1 for **reviewer EFqS**. (3) Dependence on LLMs. Like prior work, our method relies on LLMs, which may hallucinate. Potential solutions include using more advanced models or human verification. (4) Generalization scope. Our evaluation focuses on close games with clear rewards. Extending to real-world tasks (e.g., embodied AI) introduces challenges like real-time perception and physical constraints. We will discuss these limitations explicitly in the paper.
>
> **Q4: How does SemDiv handle contradictory behaviors across teammates?**
>
> A: SemDiv uses a multi-head architecture (Sec. 3.3) to mitigate catastrophic forgetting in multi-agent settings [5,6]. Testing confirms the final agents successfully coordinate with all 6 learned teammates, including those with contradictory behaviors (Table 10).
>
> We apologize for these unclear expressions, and will improve them in the paper.
>
> - Complementary policy: we aim to train a teammate policy $\pi^{tm}$ (e.g., player A in a two-player game) to form a team and train with agent $\pi^{ag}$ (e.g., the other player B). To achieve this, $\pi^{tm}$ must first learn coordination by training alongside a complementary policy $\tilde{\pi}^{tm}$ (which also controls Player B) using MARL algorithms. This training phase with $\tilde{\pi}^{tm}$ ensures that $\pi^{tm}$ develops specific coordination behaviors before interacting with $\pi^{ag}$.
> - $\lambda$ configurations in Eq. 2: The equation ensures the new teammate $\pi^{tm}_m$ differs from previous ones $\pi^{tm}_j$. When $\lambda_1 = 1, \lambda_2 = 0$, it verifies that agents trained with $\pi^{tm}_j$ cannot achieve comparable original task rewards. When $\lambda_1 = 0, \lambda_2 = 1$, it checks the same for shaped rewards. Together, these configurations strictly guarantee the new teammate's distinctiveness.
> - Explanations to the figures and SemDiv-R1/R2: Please refer to the reply for **reviewer XbFR**.
> - Second paragraph of Section 4.1: The *five* teammates mentioned here are the ones for testing different methods, rather than the *six* teammates generated during training. We again apologize for the unclear expressions.
>
> References
>
> [1] Liu et al. Maximum entropy heterogeneous-agent reinforcement learning. ICLR 2024.
>
> [2] Xie et al. Text2Reward: Reward Shaping with Language Models for Reinforcement Learning. ICLR 2024.
>
> [3] Ma et al. Eureka: Human-Level Reward Design via Coding Large Language Models. ICLR 2024.
>
> [4] Ma et al. DrEureka: Language Model Guided Sim-To-Real Transfer. RSS 2024.
>
> [5] Yuan et al. Multi-agent Continual Coordination via Progressive Task Contextualization. TNNLS 2024.
>
> [6] Yuan et al. Learning to Coordinate with Anyone. DAI 2023.

---

> > ### Comment · Reviewer_rgCD · 2025-04-04
> >
> > Thank you to the authors for their response and the additional experiments. It is good to see the improvements made to the manuscript.
> >
> > However, my concerns regarding the applicability and scalability of the proposed method persist. As the authors themselves acknowledge, the method is subject to significant constraints—such as reliance on specific APIs and language descriptions—and I do not see a clear path for overcoming these limitations. Its scalability with respect to the number of agents also appears limited, demonstrated only up to three agents. Therefore, I will maintain my score until substantial improvements are made.
> >
> > **Update**: Regarding the first concern, I understand that it may not be possible to address it through experiments currently. Nevertheless, I believe it is necessary to provide a detailed and convincing discussion of the rationale behind the currently constrained setup.

---

> > > ### Author Response · Authors · 2025-04-05
> > >
> > > We sincerely appreciate the reviewer's feedback and the opportunity to address the concerns regarding the applicability and scalability of our method. Below, we provide a detailed response to the raised issues, along with additional experimental results to support our claims.
> > >
> > > ### Applicability: Reliance on Specific APIs and Language Descriptions
> > >
> > > The reviewer raises valid concerns about dependencies on environment APIs and language descriptions. We argue these are not fundamental limitations for the following reasons:
> > >
> > > - **Environment APIs**: Our method uses APIs only to access basic information (e.g., collected food in LBF or defeated enemies in SMACv2). This aligns with standard practices in LLM-assisted RL [1], where APIs enable reward generation [2,3,4], embodied decision making [5,6,7], etc. Even without explicit APIs, lightweight functions can extract equivalent data from state representations (e.g., tracking food collections through certain dimensions) to build prompts for SemDiv.
> > > - **Language Descriptions**: The head selection module requires minimal natural language (under 10 words in our experiments). This module serves as an interpretable alternative to interaction-based teammate modeling [8,9,10] and is not central to the framework’s core mechanics. Its simplicity also facilitates future integration with human partners.
> > >
> > > ### Scalability: Number of Agents
> > >
> > > SemDiv is agnostic to team sizes. To further empirically validate scalability, we extend the SMACv2 task into a **10vs10** scenario, where SemDiv still outperforms baselines (see table below). In this scenario, our team with 10 marines needs to fight 10 enemies split into two groups. During testing, five unseen allies from another team will form a new team with five of our agents. The prompts are slightly modified based on the original two-agent version. For efficiency during rebuttal, we reduced the number of training teammate groups from 6 to 3 for all methods. These additional experiments validate the scalability of SemDiv. For even larger systems, techniques like mean-field RL [11] or hierarchical grouping [12] could be integrated to further enhance efficiency in future work.
> > >
> > > Table 1. Number of killed enemies that the testing teammates prefer to kill (mean ± std).
> > >
> > > | Testing teammates | FCP | MEP | SemDiv |
> > > | --- | --- | --- | --- |
> > > |  Attack group 1 | 3.72 ± 0.04 | 3.8 ± 0.05 | 4.45 ± 0.10 |
> > > |  Attack group 2 | 3.88 ± 0.07 | 3.98 ± 0.08 | 4.41 ± 0.08 |
> > > | Avg | 3.80 | 3.89 | 4.43 |
> > >
> > > We hope these responses alleviate the reviewer’s concerns. We are committed to addressing limitations transparently and believe the additional experiments will strengthen the paper’s contributions. If you have any further questions or suggestions, we would be more than happy to address them. We again truly appreciate your thoughtful feedback and consideration.
> > >
> > > References:
> > >
> > > [1] Cao et al. Survey on Large Language Model-Enhanced Reinforcement Learning: Concept, Taxonomy, and Methods. TNNLS 2024.
> > >
> > > [2] Xie et al. Text2Reward: Reward Shaping with Language Models for Reinforcement Learning. ICLR 2024.
> > >
> > > [3] Ma et al. Eureka: Human-Level Reward Design via Coding Large Language Models. ICLR 2024.
> > >
> > > [4] Ma et al. DrEureka: Language Model Guided Sim-To-Real Transfer. RSS 2024.
> > >
> > > [5] Zhang et al. Building Cooperative Embodied Agents Modularly with Large Language Models. ICLR 2024.
> > >
> > > [6] Du et al. Constrained Human-AI Cooperation: An Inclusive Embodied Social Intelligence Challenge. NeurIPS 2024.
> > >
> > > [7] Chang et al. PARTNR: A Benchmark for Planning and Reasoning in Embodied Multi-agent Tasks. ICLR 2025.
> > >
> > > [8] Zhang et al. Fast Teammate Adaptation in the Presence of Sudden Policy Change. UAI 2023.
> > >
> > > [9] Yuan et al. Learning to Coordinate with Anyone. DAI 2023.
> > >
> > > [10] Ma et al. Fast Peer Adaptation with Context-aware Exploration. ICML 2024.
> > >
> > > [11] Yang et al. Mean Field Multi-Agent Reinforcement Learning. ICML 2018.
> > >
> > > [12] Christianos et al. Scaling Multi-Agent Reinforcement Learning with Selective Parameter Sharing. ICML 2021.

---

### Official Review · Reviewer_XbFR · 2025-03-11

**Overall Recommendation:** 3

**Summary:**

This paper proposes a teammate generation method called SemDiv, which uses LLMs to learn diverse coordination behaviors a the “semantic” level. SemDiv generates novel teammates by iterating the following steps: (1) generating natural language description of a novel coordination behavior, (2) translating it into a shaping reward function for training a teammate policy, (3) verifying whether training using such a reward function generates a meaningfully different policy, (4) if verified, train a new policy head to coordinate with the newly generated teammate, using a continual learning objective to avoid forgetting how to coordinate with previously seen teammates. At test time with an unknown teammate and a behavior description of that teammate, SemDiv uses an LLM to select the best response policy head.

The method is evaluated on 4 MARL environments: LBF, Predator-Prey, SMAC-v2, and Google Research Football (GRF), with 5 unseen teammates per task. Emprical results show that SemDiv outperforms various teammate generation baselines and LLM baselines in terms of generalization.

**Claims And Evidence:**

Most claims are somewhat supported by evidence. There are some claims that are misleading or insufficiently supported, which I have pointed out below.

- “First, the exploration of the teammate policy space is inefficient, as teammates are driven to optimize for differences at the policy-level rather than actively discovering novel coordination behaviors at the semantic level.” ([pdf](zotero://open-pdf/library/items/B38DLBJN?page=1))

    - The authors do not provide any theoretical justification or empirical demonstration of why optimizing policy level differences is bad. It’s not clear what the difference between “policy level” vs “semantic level” is.

- A key claim of this paper is that LLMs are better able to generate diverse teammates than non-LLM teammate generation baselines

    - I don’t think the paper presents enough evidence for this claim.

    - The case study figure where the strategies generated by SemDiv and FCP is good (Fig 3c), but not totally convincing, because (1) FCP is one of the weakest teammate generation baselines, and (2) T-SNE can produce different results when run with different seeds. The evidence would be much stronger if the authors do two things: first, re-generate Fig. 3c for the next-strongest non-SemDiv baseline for all tasks, and second, provide cross-play matrices for the populations generated by all teammate generation methods.

- Another key claim of this paper is that the proposed LLM-based teammate generation process generalizes better to unseen teammates, compared to non-LLM based methods.

    - Evidence provided in form of performance comparisons against basedline non-LLM methods (FCP, MEP, LIPO, Macop) and an LLM-agent only baseline, where manually designed unseen teammates are used to test SemDiv and baseline

    - However, given the external calls to LLMs and potentially computationally expensive verification sub-routines within SemDiv (e.g. testing if the reward function is valid), the authors should also report the wall-clock time required to train each method to obtain the results in Table 1.

- Misleading claim - the abstract states that SemDiv is evaluated on 20 unseen teammates and 4 tasks, making it unclear whether it was evaluated using 20 teammates per task, or divided evenly among the tasks (as turns out to be the case). The authors should reword that claim in the abstract (and anywhere else it appears in the paper) to avoid implying that their method was evaluated with more teammates than it actualy was.

- Misleading characterization the paper is couched in terms of N-agent teams, and in various places refers to the number of controlled agents as {1, …, n_ag} and the number of uncontrolled as {1, …, n_tm}. This implies that N>2. While I’m fine with describing the method in greater generality, the paper should clearly state somewhere that N=2 in all evaluation settings.

- Misleading characterization - SemDiv’s continual ego agent learning procedure and multi-head architecture directly comes from Macop (Yuan et al. 2023). This is fine, as this isn’t really the main contribution of the paper, but Section 3.3 (which describes the ego agent learning procedure) should clearly acknowledge this. and make it clear what is different in this work.

**Essential References Not Discussed:**

No

**Experimental Designs Or Analyses:**

See above.

**Methods And Evaluation Criteria:**

Overall, the proposed method makes sense, and the experiments evaluate the key claims of the method (improved generalization + improved teammate diversity). I have some concerns with the evidence provided for the claims, which I have described previously. Some additional concerns about the method are provided below.

- The LLM is used to generate/verify several key points in the teammate generation procedure. i would like to see evaluations of each component in terms of metrics such as the number of attempts, success rate, and wall-clock time

    - Generating reward function programs

    - Verification of alignment between behaviors/policies

    - Policy selection process for testing with unknown teammate.

- Statistical significance measures: results are presented with mean +/- std deviations throughout the paper, which reflects the variance in the mean performance for each method. To provide information about the statistical significance of the results, the authors should provide statistical significance tests comparing the performance of SemDiv to the next best performing method, or computing 95% CI’s instead.

**Other Comments Or Suggestions:**

- Details that need to be made clearer:

    - What percentage of the time does SemDiv generate valid vs invalid teammates?

    - Reading the methods section was confusing, because I thought that the paper addressed the scenario of N>=2. The authors should specify that the paper addresses N=2.

- Each figure caption should explain the main message that the figure is meant to show, to support readers skimming the paper based on figures alone

- There are a couple issues with Table 1:

    - Since {SemDiv, Macop}-R1, R2 are upper bounds, all four methods should be included at the top, perhaps in the oracle section, or perhaps in their own section. It’s confusing to have it presented together with SemDiv.

    - Shouldn’t Macop-R1, R2 be grayed out too, since it is also an upper bound and presumably excluded from consideration for computing the best result?

**Other Strengths And Weaknesses:**

- Strengths:

    - Overall, the paper is well-written and well-situated in the field of ad hoc teamwork

    - The idea is novel and timely, and the overall framework is quite compelling.

    - The results convincingly show that SemDiv outperforms other common teammate generation baselines.

- Weaknesses

    - SemDiv requires natural language behavior descriptions from unknown teammates to allow an LLM to select the optimal head.

        - In my view, this is the largest weakness of the paper. SemDIV is presented as an end-to-end teammate generation + AHT method, but this is a pretty large limitation of the AHT component of the method. The policy selection problem makes up a large portion of the challenge for AHT methods (e.g. PLASTIC by Barret et al. 2017). Assuming access to a natural language description of the unknown teammate essentially assumes an oracle for the teammate modeling problem (i.e. generating a characterization of an unseen teammate, and relating it to the representation of known teammates).

        - However, I believe that the main contribution of the method is showing how to generate teammates using LLMs/natural language descriptions, which is sufficient on its own, even without any novelties in the ego agent learning process. The authors should explicitly acknowledge this limitation.

    - Multiple misleading claims are made.


Barrett, Samuel, Avi Rosenfeld, Sarit Kraus, and Peter Stone. 2017. “Making Friends on the Fly: Cooperating with New Teammates.” *Artificial Intelligence* 242 (January):132–71. [https://doi.org/10.1016/j.artint.2016.10.005](https://doi.org/10.1016/j.artint.2016.10.005).

**Questions For Authors:**

1. In Table 1, the difference between SemDiv and SemDiv-PBT is much larger than the gap between SemDiv-PBT and non-LLM teammate generation methods (e.g. Macop-PBT, FCP, MEP, LIPO), which makes me wonder if SemDiv is performing better than the non-LLM teammate generation methods because of the continual learning paradigm introduced by MACOP, rather than because of the generated teammates. Can you present the performance of MACOP (the original, non-PBT version) as well, in Table 1?

2. [Clarification Question] After reading the paper, my impression is that SemDiv requires behavior descriptions from unknown teammates. However, the experiments section (L242-245) states that the teammates’ behavior descriptions remain unknown to the tested methods during training. Does SemDiv select a policy head without the behavior description? If so, then what is the purpose of the Semdiv-Dist baseline?

3. Given that the SemDiv does require behavior descriptions:

    1. Can the authors provide a discussion of how reasonable this assumption is?

    2. Semdiv-Dist performs much worse than Semdiv, suggesting that the behavior descriptions do a lot of the ‘heavy lifting’ for enabling generalization to unseen teammates. Do the authors have any thoughts on how to improve this?

4. Reward shaping: the criteria outlined for whether the reward function is valid seems to require training a policy under the reward function  as an inner-loop step (Line 173 in the submitted PDF). Can the authors comment on how many timesteps are allocated to train the teammate under hte reward function? Some reward functions can only be optimized in a large number of steps -- how does this affect the “deepness” of the generated behaviors? Is it the case that only “easy/simple” behaviors can be generated?

**Relation To Broader Scientific Literature:**

This method is a teammate generation method for ad hoc teamwork/zero-shot coordination. The authors propose using LLMs to generate novel teammate behaviors, which to my knowledge, has not been considered/addressed before for ad hoc teamwork.

**Theoretical Claims:**

N/A

---

> ### Author Rebuttal · Authors · 2025-04-01
>
> We sincerely thank you for the valuable comments!
>
> **Q1: Difference between “policy level” and “semantic level”, and why optimizing policy level differences is bad.**
>
> A: Policy-level methods mainly optimize policy-space diversity without explicitly modeling the corresponding semantics. In this approach, the search space grows exponentially with the number of agents, leading to inefficient exploration. In contrast, our semantic-level method explicitly models human-interpretable coordination behaviors through language-guided generation. By clustering similar policies into semantically meaningful behaviors, our approach significantly reduces the complexity of the search space.
>
> Prior works [1,2] have provided analysis and conducted experiments to demonstrate that traditional policy-level methods (e.g., FCP and LIPO) struggle to efficiently explore teammate policy spaces. We further validate this claim through comprehensive performance evaluations and trajectory analysis, highlighting the advantages of semantic-level optimization over policy-level approaches.
>
> **Q2: More results on t-SNE visualization, cross-play matrices, etc.**
>
> A: We report these important results in this [link](https://telling-floor-898.notion.site/1c7c2fed721a80b9ba7ef7fa2b3bffed), and will add them in the paper.
>
> **Q3: Requirement of natural language behavior descriptions from unknown teammates.**
>
> A: Please refer to Q1 for **reviewer EFqS**.
>
> **Q4: The performance of Macop.**
>
> A: We would like to clarify that we have actually included the performance of Macop in Table 1, though presented as its *upper bound* versions that directly report the results of the best policy heads. However, there is still a gap between Macop-{R1, R2} and SemDiv, proving the impact of SemDiv’s semantically diverse teammates.
>
> **Q5: [Clarification Question] How SemDiv selects policy heads and the SemDiv-Dist baseline.**
>
> A: During training, all methods are unaware of testing teammates' behavior descriptions. SemDiv agents only select heads for the corresponding training teammates for policy verification. During testing, SemDiv agents select heads according to the testing teammates’ behavior descriptions. The SemDiv-Dist baseline also requires these descriptions during testing to select heads. Instead of using LLMs for reasoning, it selects the head that minimizes Distance(embed($b$), embed($b_{test}$)), where $b$ is the behavior learned by the head, $b_{test}$ is the behavior of the testing teammate. To improve SemDiv-Dist, we can finetune the language model that computes the embeddings with task-specific data, enhancing its understanding of the task.
>
> **Q6: The reward shaping problem.**
>
> A: In our work, we empirically set a fixed step limit for inner-loop training, a practical trade-off that may limit the "deepness" of learnable behaviors. While this suffices for simpler tasks, we fully acknowledge the challenge for more complex behaviors. Future work will adopt Neural Architecture Search (NAS) [3] to automate hyperparameter setting, enabling efficient training of deeper behaviors under shaped rewards. We will clarify this limitation and solution in the paper.
>
> We apologize for these unclear expressions, and will improve them in the paper.
>
> - Number of testing teammates and controlled agents: We have modified the Abstract and Introduction to explicitly state: “Evaluation with five unseen representative teammates per environment”, and we will add an explanation in the Problem Formulation that we focus on scenarios where N=2.
> - Related work of Section 3.3: Please refer to the reply for **reviewer Q2F9**.
> - More detailed figure captions:
>     - Figure 1: An overview of the training and testing process of SemDiv. Left: During training, SemDiv proposes novel coordination behaviors in natural language and transform them into teammate policies for agent learning. Right: During testing, SemDiv takes as input the description of the unseen teammates and selects the optimal learned policy for coordination.
>     - Figure 2: The overall workflow of SemDiv. (a) Generating coordination behavior. SemDiv iteratively generates of semantically diverse coordination behaviors, enabling efficient exploration of the teammate policy space. (b) Training aligned teammate policy. For each coordination behavior described in natural language, a teammate policy is trained to align with that behavior. (c) Training agents. Agents are continually trained with these teammates, developing strong coordination ability.
> - Upper bounds: {SemDiv, Macop}-R1, R2 are only upper bounds of {SemDiv, Macop}, which directly report the results of the best policy heads to investigate the impact of the head selection module. We again apologize for these unclear expressions.
>
> References
>
> [1] Yuan et al. Learning to Coordinate with Anyone. DAI 2023.
>
> [2] Rahman et al. Minimum Coverage Sets for Training Robust Ad Hoc Teamwork Agents. AAAI 2024.
>
> [3] Elsken et al. Neural Architecture Search: A Survey. JMLR 2019.

---

### Official Review · Reviewer_EFqS · 2025-03-13

**Overall Recommendation:** 3

**Summary:**

The paper proposes a novel partner generation method, which iteratively generates new partner through generated reward functions via LLM queries. The input to the LLM contains semantic information of the behavior, allowing the method to include semantic information for generating the partners in addition to policy-level validity and diversity check. The proposed method also leverages the LLM for selecting the right best response policy given a behavior description of the test partner. The method outperforms various baselines across diverse environments.

**Claims And Evidence:**

> This demonstrates that generating semantically diverse teammates not only enables more efficient exploration of the teammate policy space but also facilitates the discovery of coordination behaviors that policy-level exploration alone cannot cover.

The paper claims that "policy-level exploration" cannot discover certain behaviors. But in the experiment, it only uses FCP---which does not even explicitly generate diverse partners---as the only reference for this claim. It is not fair to claim this without testing more sophisticated "policy-level exploration" methods.

**Essential References Not Discussed:**

The paper should mention the work that Eq. 2 is based on ([1]) and why using Eq. 2 could possibly check for policy similarity in the first place. In its current form, the paper simply mentions that it uses Eq. 2  to verify diversity without giving any intuition. The paper already has the citation in various places but not at Eq. 2.

[1] Charakorn et al. "Generating diverse cooperative agents by learning incompatible policies." The Eleventh International Conference on Learning Representations. 2023.

**Experimental Designs Or Analyses:**

The experiments and analyses are reasonable with statistical confidence.

**Methods And Evaluation Criteria:**

The method and evaluation are reasonable. Though the paper would be more convincing with an ablation study on the Policy Verification is included (see Comments or Suggestions)

**Other Comments Or Suggestions:**

### Comments
> First, the exploration of the teammate policy space is inefficient, as teammates are driven to optimize for differences at the policy-level rather than actively discovering novel coordination behaviors at the semantic-level.

To strengthen this statement, the paper should substantiate why "the exploration of the teammate policy space is inefficient". Is there any existing paper that confirm this statement?

> As we aim to study teammates generation and agents coordination at the semantic-level, we consider scenarios in which the group of teammates $\pi^\text{tm}$ provides a natural language description $b$ prior to the execution phase.

The reason given for the use of prior communication is not sound. One could also communicate without semantic-level coordination. Coordinating at the semantic-level could also omit communication.

### Suggestions
- The baselines do not use the "Policy Verification". Since it is very likely that with high regularization, XP-min methods tend to generate incapable policies, just like non-functional reward functions generated by the LLM. The paper would benefit by a baseline that uses the Policy Verification as well. This would confirm that the semantic information is indeed useful for partner generation, not the Policy Verification part. In its current form, there could be an alternate explanation that the Policy Verification does the heavy lifting of the improvement.

- Related to Policy Verification, I wonder how much percentage of the generated reward functions are invalid? How much percentage of them leads to executable reward functions but does not pass the Policy Verification? What is the average number of repetitions before generating a valid policy?

I'm willing to increase the score if all these concerns are addressed properly.

**Other Strengths And Weaknesses:**

Strengths
- The idea of using LLM for partner generation is novel.
- The evaluation includes many SOTA baselines.

Weaknesses
- The claim in Section 4.4 is not fair (see Claims and Evidence).
- It is not clear whether the use of the LLM is the main contributor to the improvement over the baseline (see Suggestions).
- Some information about the pipeline is not provided fully, e.g., how `info_sim` or agent behaviors are represented for the LLM.

**Questions For Authors:**

- How is the `info_sim` represented?

- How are agent behaviors represented in text before feeding into the LLM (for the alignment step)? I suppose different environments would have different behavior representations/descriptions.

- Is the better performance stemmed from the fact that XP-min based methods are unstable/inefficient? My understanding is that the proposed method still generates incompatible policies but through generated reward functions (with the "diversity check" which checks the compatibility of newly generated policy) instead directly optimizing the "diversity check", which is based on the XP-min methods.

**Relation To Broader Scientific Literature:**

The paper could be impactful as it introduces the use of LLMs for generating diverse partners. Prior work focuses on generating diverse partners at the policy-level. This approach incorporates semantic information to generate more diverse partners.

**Theoretical Claims:**

N/A

---

> ### Author Rebuttal · Authors · 2025-04-01
>
> We sincerely thank you for the valuable comments! Extra experimental results can be found in this [link](https://telling-floor-898.notion.site/1c7c2fed721a80b9ba7ef7fa2b3bffed).
>
> **Q1: The motivation for unseen teammate communicating behaviors before testing.**
>
> A: Communication and intention sharing are the basic and core techniques in multi-agent systems [1,2], mitigating partial observability. Following these techniques, our approach leverages one-time natural language communication to boost coordination efficiency. For situations where communication is entirely infeasible, SemDiv can still infer teammate behaviors from interactions using techniques in [3,4]. We note this as a future direction to extend our framework to zero-comm environments.
>
> **Q2: Compare more baselines to support the claims in Section 4.4.**
>
> A: As shown in Table 1 in the link, even with extra policy-level diversity techniques and larger population sizes, policy-level baselines still achieve limited performance, further indicating the necessity and efficiency of semantic-level diversity.
>
> **Q3: Compare baselines with Policy Verification.**
>
> A: Even with Policy Verification, population-based baselines show limited improvement (Table 2). The Policy Verification process: (1) Train 12 teammates, verifying all can complete the task. (2) Select the 6 most diverse policies:
>
> - MEP: Farthest from population mean.
> - LIPO: Iteratively pick worst-performing partners.
>
> No behavior verification is needed as baselines lack semantic info. Results confirm LLM-assisted diversity drives SemDiv's success, not just Policy Verification.
>
> **Q4: Pass rate of Policy Verification in SemDiv.**
>
> A: We report the results of Policy Verification (Table 3), including (1) the number of occurrences of different issues averaged over 3 random seeds and (2) total pass rate. SemDiv efficiently generates diverse teammates while verifying their quality. We will add these results in the paper.
>
> **Q5: Similarity information and agent behavior text representation.**
>
> A: We simply represent info_sim as: “Not-novel example: {behavior}. This behavior is the same as {similar_behavior}“. For behavior texts, we write light-weight functions to extract information from trajectory data and fill in text templates, aligning with previous works [5,6]. For example, in SMACv2, info = “Agents killed enemy {enemy_id} …”. We will add more details of text representation in the paper.
>
> **Q6: Whether SemDiv’s improvement is stemmed from the instability of baselines.**
>
> A: SemDiv’s improvement is stemmed from better semantic-level diversity rather than exploting the instability of baselines. We report the performance of all generated teammates in Table 4. The teammate stability and quality of baselines is similar to that of SemDiv.
>
> **Q7: Substantiate previous methods‘ inefficient exploration of the teammate policy space.**
>
> Efficiently exploring diverse high-quality teammate policies is a fundamental challenge in MARL for open environments [7]. Early methods primarily imposed regularization at the action level, requiring exhaustive traversal of the joint policy space, which becomes computationally prohibitive in complex scenarios. Recent works like [4,8] have demonstrated the limitations of classic policy-level methods like FCP and LIPO. In contrast, our approach abstracts the joint policy space into a higher-level semantic space, where a single semantic behavior can correspond to a wide range of policies. This abstraction explicitly reduces exploration difficulty by decoupling diversity from low-level action redundancy. Empirical results validate that SemDiv achieves more efficient and scalable policy exploration compared to prior works, representing a significant technical advancement in this direction.
>
> **Q8: Mention related work in Eq. 2.**
>
> A: We apologize for not properly citing the work [9] when introducing Eq. 2 and explaining its relevance for policy similarity verification. We will correct this by adding the citation at Eq. 2 and providing a clearer discussion.
>
> References
>
> [1] Albrecht et al. Multi-Agent Reinforcement Learning: Foundations and Modern Approaches. MIT Press, 2024.
>
> [2] Zhu et al. A survey of multi-agent deep reinforcement learning with communication. Auton. Agents Multi Agent Syst. 38(1): 4 (2024).
>
> [3] Yuan et al. Multi-agent Continual Coordination via Progressive Task Contextualization. TNNLS 2024.
>
> [4] Yuan et al. Learning to Coordinate with Anyone. DAI 2023.
>
> [5] Xie et al. Text2Reward: Reward Shaping with Language Models for Reinforcement Learning. ICLR 2024.
>
> [6] Ma et al. Eureka: Human-Level Reward Design via Coding Large Language Models. ICLR 2024.
>
> [7] Yuan et al. A Survey of Progress on Cooperative Multi-agent Reinforcement Learning in Open Environment. 2023.
>
> [8] Rahman et al. Minimum Coverage Sets for Training Robust Ad Hoc Teamwork Agents. AAAI 2024.
>
> [9] Charakorn et al. Generating diverse cooperative agents by learning incompatible policies. ICLR 2023.

---

> > ### Comment · Reviewer_EFqS · 2025-04-02
> >
> > Thank you the authors for providing a thorough response with additional experiments.
> >
> > > Q1: The motivation for unseen teammate communicating behaviors before testing.
> >
> > I do agree that using communication makes sense and can help adaptation of cooperative agents. My comment was specifically how the use of communication is motivated in the text. I suggest the authors clarify the motivation in the main text in the revision.
> >
> > > Q2: Compare more baselines to support the claims in Section 4.4.
> >
> > Thank you for providing this result. I believe Section 4.4 is now much more convincing.
> >
> > > Q3: Compare baselines with Policy Verification.
> >
> > Thank you for conducting this experiment. This result is another crucial ablation. I suggest the authors include this in the paper.
> >
> > > Q4: Pass rate of Policy Verification in SemDiv.
> >
> > Thank you for providing more details on how SemDiv works.
> >
> > > Q5: Similarity information and agent behavior text representation.
> >
> > Thank you. Please add these details to the paper.
> >
> > > Q6: Whether SemDiv’s improvement is stemmed from the instability of baselines.
> >
> > Thank you for the clarification.
> >
> > > Q7: Substantiate previous methods‘ inefficient exploration of the teammate policy space.
> >
> > Please add this motivation to the paper.
> >
> > > Q8: Mention related work in Eq. 2.
> >
> > Please add the explanation to the paper.
> >
> > My concerns are largely addressed. I raise my score from 2 to 3.

---

> > > ### Author Response · Authors · 2025-04-03
> > >
> > > Thank you for taking the time to review our paper and for raising your score, we sincerely appreciate it.
> > >
> > > Regarding your comment on the motivation for using communication, we agree that this point could be more clearly articulated. In the revised version, we will clarify that our motivation stems from a key limitation in current teammate adaptation approaches [1,2,3]: they typically require interaction data with the test-time teammates, which can be costly or infeasible in complex environments. To address this challenge, we propose leveraging communication as a means of intention sharing, thereby reducing the need for extensive adaptation. Moreover, with the rapid advancement of LLMs, communication can now be easily realized through natural language, which not only improves interpretability but also opens up new possibilities for coordination with real human partners.
> > >
> > > If you have any further questions or suggestions, we would be more than happy to address them. We truly appreciate your thoughtful feedback and consideration.
> > >
> > > References
> > >
> > > [1] Zhang et al. Fast Teammate Adaptation in the Presence of Sudden Policy Change. UAI 2023.
> > >
> > > [2] Yuan et al. Learning to Coordinate with Anyone. DAI 2023.
> > >
> > > [3] Ma et al. Fast Peer Adaptation with Context-aware Exploration. ICML 2024.

---

### Decision · Program_Chairs · 2025-05-01

**Decision:**

Accept (poster)

**Comment:**

After discussion, reviewers have reached a consensus that the paper should be accepted, arriving at scores of [3,3,5,3].

Reviewer Q2F9 was the strongest advocate for the paper, and explained that: "Most existing generalizable techniques (based on trajectory/policy diversity or cross-play minimization) often result in arbitrary conventions, none of which are aligned to human intuitions. By introducing LLMs for convention formation, we now have more aligned conventions while still maintaining diversity (as verified by the cross-play alignment). This whole pipeline has the potential to significantly improve the field of human-AI collaboration".

Reviewer rgCD was the most hesitant, and while they did not feel strongly that the paper should be rejected, they had questions about how broadly applicable the method could be, and hoped that the authors would "engage more deeply with the potential of leveraging LLMs or explore ways to address some of the constraints introduced by their use."

In summary, the reviewers agree that the proposed method is novel and promising, and that the paper provides strong empirical results. However, several questioned a critical drawback of the method, which is that it assumes access to a natural language description of the teammate at test time, which may be unrealistic for real-world settings.

In the rebuttal, the authors commented on this weakness, stating "Communication and intention sharing are the basic and core techniques in multi-agent systems [1,2], mitigating partial observability. Following these techniques, our approach leverages one-time natural language communication to boost coordination efficiency. For situations where communication is entirely infeasible, SemDiv can still infer teammate behaviors from interactions using techniques in [3,4]. We note this as a future direction to extend our framework to zero-comm environments."

As AC, I still have questions about the realism of this assumption, but agree that the method for generating training teammates is interesting and novel. Given this evidence, I am recommending accept.